# Control of motor landing and processivity by the CAP-Gly domain in the KIF13B tail

Xiangyu Fan [1] & Richard J. McKenney [1] ✉

Microtubules are major components of the eukaryotic cytoskeleton. Post-translational modifications (PTMs) of tubulin regulates interactions with microtubule-associated proteins (MAPs). One unique PTM is the cyclical removal and re-addition of the C-terminal tyrosine of α-tubulin and MAPs containing CAP-Gly domains specifically recognize tyrosinated microtubules. KIF13B, a long-distance transport kinesin, contains a conserved CAP-Gly domain, but the role of the CAP-Gly domain in KIF13B's motility along microtubules remains unknown. To address this, we investigate the interaction between KIF13B's CAP-Gly domain, and tyrosinated microtubules. We find that KIF13B's CAP-Gly domain influences the initial motor-microtubule interaction, as well as processive motility along microtubules. The effect of the CAP-Gly domain is enhanced when the motor domain is in the ADP state, suggesting an interplay between the N-terminal motor domain and C-terminal CAP-Gly domain. These results reveal that specialized kinesin tail domains play active roles in the initiation and continuation of motor movement.

Microtubules are a critical component of the eukaryotic cytoskeleton and are essential for various cellular functions, such as cell division and motility, intracellular signaling and transport, cell differentiation, and generation of specific organelles[1]. Microtubules are polymers of tubulin dimers, themselves composed of various α- and β-tubulin isotypes[2]. In addition to tubulin isotypes, the cytoplasmic surface of the microtubule polymer is modified with a variety of post-translational modifications (PTMs) to the C-terminal tail domains of the tubulin dimers. The large variety of tubulin isotypes and PTMs is collectively referred to as the "tubulin code"[1,3,4]. Cells express enzymes that act as "writers" of the tubulin code and modify the surface of the microtubule in a spatiotemporally controlled manner[4,5]. In addition, a large repertoire of microtubule-associated proteins (MAPs), including molecular motors, act as "readers" of the tubulin code. One example is the direct effect of tubulin polyglutamylation on the recruitment and severing activities of spastin and katanin[6–8]. While it is hypothesized that the intracellular localization and activities of molecular motors could be directly modulated by tubulin PTMs, abundant evidence for this idea remains elusive because of the complex environment of the cytoplasm and the relatively weak effects of tubulin PTMs on motor activity in vitro[9,10].

One unique PTM on microtubules is the enzyme-driven tubulin detyrosination/tyrosination cycle, in which the encoded C-terminal tyrosine residue of mammalian α-tubulin can be removed from the polypeptide chain by the tubulin carboxypeptidase vasohibins (VASHs) with the cooperation of its cofactor, small vasohibin-binding protein (SVBP)[11,12]. Tyrosine can also be added back onto α-tubulin by a tubulin tyrosine ligase (TTL)[13,14]. In most cell types, tyrosinated microtubules make up the bulk of the polymer mass and are typically more dynamic than detyrosinated microtubules[15,16]. Tyrosinated and detyrosinated microtubules compose distinct subsets of microtubules that orient in opposite directions within neuronal compartments[17]. Detyrosinated microtubules are more stable than tyrosinated microtubules, and cells polarize detyrosinated microtubules towards the direction of cell migration[18], suggesting distinct functionalities for the two populations of modified microtubule polymer. Thus, the detyrosination/tyrosination cycle specifies subpopulations of microtubules for cellular functions, presumably through direct effects on microtubule effectors.

Some MAPs contain an evolutionary conserved CAP-Gly (cytoskeleton-associated protein glycine-rich) domain that specifically recognizes tyrosinated microtubules[19]. Structural work has revealed that the CAP-Gly domain interacts with the C-terminal EEY/F sequence

[1]Department of Molecular and Cellular Biology, University of California - Davis, 145 Briggs Hall, Davis, CA 95616, USA. ✉e-mail: rjmckenney@ucdavis.edu

motifs of α-tubulin[20,21], or a highly similar region found in the microtubule end-binding (EB) protein family and cytoplasmic linker protein (CLIP) family[20,22–24]. Some CAP-Gly domain-containing proteins, such as CLIP-170 and CLIP-115, participate in the regulation of microtubule dynamics and the recruitment of proteins to the microtubule plus-end[25,26]. Tubulin tyrosination can also affect the targeting and movement of motor proteins along microtubules. The p150[Glued] (hereafter p150) subunit of the dynactin complex also contains a CAP-Gly domain that is important for targeting the dynactin complex to microtubule plus-ends in cells and initiating retrograde dynein movement from these regions[27,28]. In vitro, the p150 CAP-Gly domain strongly biases the landing of assembled dynein-dynactin-cargo adapter complexes onto tyrosinated, versus detyrosinated microtubules[29,30], but it is not required for sustained processive motility of these complexes after initiation of movement[29]. The p150 CAP-Gly domain also enhances the dynein–dynactin interaction with microtubules under load[31]. Therefore, the p150 CAP-Gly domain directly affects the landing and force production properties of the cytoplasmic dynein–dynactin complex, perhaps representing the largest effect of a tubulin PTM on molecular motor motility measured to date.

Within the kinesin family, the microtubule depolymerizing activity of kinesin-13 (MCAK) is directly regulated by tubulin tyrosination[10,32]. Tyrosination also affects the velocity and processivity of kinesin-2 family motors[10]. Interestingly, within the large kinesin-3 family, the KIF13B motor contains a predicted CAP-Gly domain located at the C-terminus of the motor. This domain is highly conserved in all organisms with an identified KIF13B homolog (Supplementary Fig. 1A). KIF13B is involved in various cellular processes, such as axon formation, neuronal cell polarity, regulation of angiogenesis, directional migration of the primordial germ cell in *Xenopus* embryos, the uptake of lipoproteins via endocytosis, and the transport of Rab6 secretory vesicles from the Golgi[33–45]. More recently, KIF13B has also been observed to move bi-directionally within mammalian primary cilia[46]. The role of KIF13B's CAP-Gly domain in these processes is not clear, but deletion of the domain in mice results in the mislocalization of the truncated motor and KIF13B cargo within cells[38].

Here, we characterized the role of the KIF13B CAP-Gly domain during its processive motion along microtubules using in vitro reconstitution with single-molecule imaging. We found that the CAP-Gly domain directly affects KIF13B's motility by increasing the landing rate and extending the run length of KIF13B along tyrosinated microtubules. Our findings reveal that specialized tail domains of kinesin motors play active roles in the initiation and continuation of motor movement during cargo transport.

## Results
### Characterization of recombinant KIF13B
We expressed and purified full-length and truncated versions of the human KIF13B motor from insect cells (Fig. 1A). All constructs were purified via a C-terminal superfolder-GFP (sfGFP)-strepII tag and gel filtration to near homogeneity (Fig. 1B). In addition to the wild-type full-length motor which is expected to exist in an autoinhibited state[47], we generated two single point mutants, KIF13B[V178Q] and KIF13B[K414A], that have been previously shown to disrupt the autoinhibition mechanism of KIF13B[48]. Additionally, we generated constructs containing only the CAP-Gly domain (KIF13[CG]), or lacking a large portion of the C-terminal tail domain (KIF13[Δtail], Fig. 1A, B). Previous studies have found that tail-truncated KIF13B motors are monomeric and must dimerize to induce processive motion along microtubules[48,49], although the inclusion of longer segments of the tail domain appears to result in an active dimeric motor[35,44]. However, photobleaching studies suggested the full-length KIF13B in cell lysates was largely monomeric[49]. To examine the oligomeric state of our KIF13B preparations, we utilized mass photometry[50] across a range of nanomolar concentrations relevant to our single-molecule assays. We observed

that the WT KIF13B motor was predominantly monomeric across a 20-fold concentration range, but we could detect a minor proportion of dimeric motors at the intermediate and high concentrations examined (Fig. 1A–C). The V178Q and K414A mutations have previously been shown to disrupt the intramolecular autoinhibition of KIF13B, which altered the monomer–dimer equilibrium for tail-truncated KIF13B[48]. Consistently, we observed our full-length KIF13B[V178Q] or KIF13B[K414A] motor preparations contained a larger proportion of dimeric motors across all concentrations tested as compared to the WT construct (Fig. 1C). Therefore, we conclude that purified, full-length KIF13B exists predominantly as a monomeric motor in solution and has a weak tendency to dimerize at higher concentrations. This result is broadly consistent with experiments performed in cell lysates[49]. Dimerization of the motor is enhanced by disruption of the autoinhibition mechanism that operates through the interaction between the motor domain and the first coiled-coil region of the motor[48].

We also examined the oligomeric state of our truncated motor constructs. We truncated KIF13B after the predicted second coiled-coil, and within the membrane-associated guanylate kinase (MAGUK) binding stalk (MBS) domain[51] to generate KIF13B[Δtail] (Fig. 1A–C). At lower concentrations, KIF13B[Δtail], existed predominantly as a monomer with a smaller population of dimeric motors. However, raising the concentration 5-fold resulted in near complete dimerization of this construct, revealing that removal of the C-terminal tail domain of KIF13B facilitates a much more dramatic concentration-dependent dimerization of the motor (Fig. 1A–C), as compared to the full-length protein. We were unable to record data for KIF13B[Δtail] at concentrations higher than 25 nM due to the overly high binding density of this protein to the coverslip surface. From these data, we conclude that the C-terminal tail region of KIF13B negatively impacts the dimerization of the motor, suggesting that molecular interactions within this region could modulate the oligomeric state of the motor in vivo. A recent structure of an autoinhibited and monomeric kinesin-3 member, KLP-6, reveals extensive interdomain interactions within the C-terminal tail of this kinesin-3 family member supporting this idea[52].

Truncation of the N-terminus produced a fragment encompassing the CAP-Gly domain and an ~80 residue unstructured C-terminal region (KIF13B[CG], Fig. 1A, Supplementary Fig. 1B). KIF13B[CG] was predominantly monomeric at the low and intermediate concentrations examined (Fig. 1A–C), while at higher concentration, KIF13B[CG] unexpectedly shifted to a dimeric form (Fig. 1C). The reason for the dimerization of KIF13B[CG] at higher concentration is unclear, but we hypothesize that the unstructured region following the CAP-Gly domain may play a role. We also note that this region is rich in basic amino acids (Supplementary Fig. 1B) which may facilitate interactions with the acidic tubulin C-terminal tail domains.

### KIF13B binds preferentially to tyrosinated microtubules
Because the tyrosination state of the microtubule affects the interactions of various motor proteins with microtubules[10,17,29,53], we sought to investigate whether the KIF13B CAP-Gly domain plays a direct role in the motor's interaction with the microtubule lattice. In principle, KIF13B contains two distinct types of microtubule-binding domains, the N-terminal ATP-dependent kinesin motor domain, and the C-terminal nucleotide-independent CAP-Gly domain (Figs. 1A, 2A). The relative role of each of these domains in the overall motor-microtubule interaction is unknown. To examine this, we directly compared the interactions of our KIF13B constructs with tyrosinated versus detyrosinated microtubules in vitro (Fig. 2A, B). In this assay, microtubules were polymerized from purified porcine brain tubulin, which is approximately 50% tyrosinated[54]. Carboxypeptidase A (CPA) treatment was used to remove the C-terminal tyrosine of α-tubulin[29]. CPA treatment strongly reduced the binding of p150 to microtubules (Supplementary Fig. 3A) as previously reported[29], confirming detyrosination of the lattice. Differentially labeled WT (tyrosinated) and

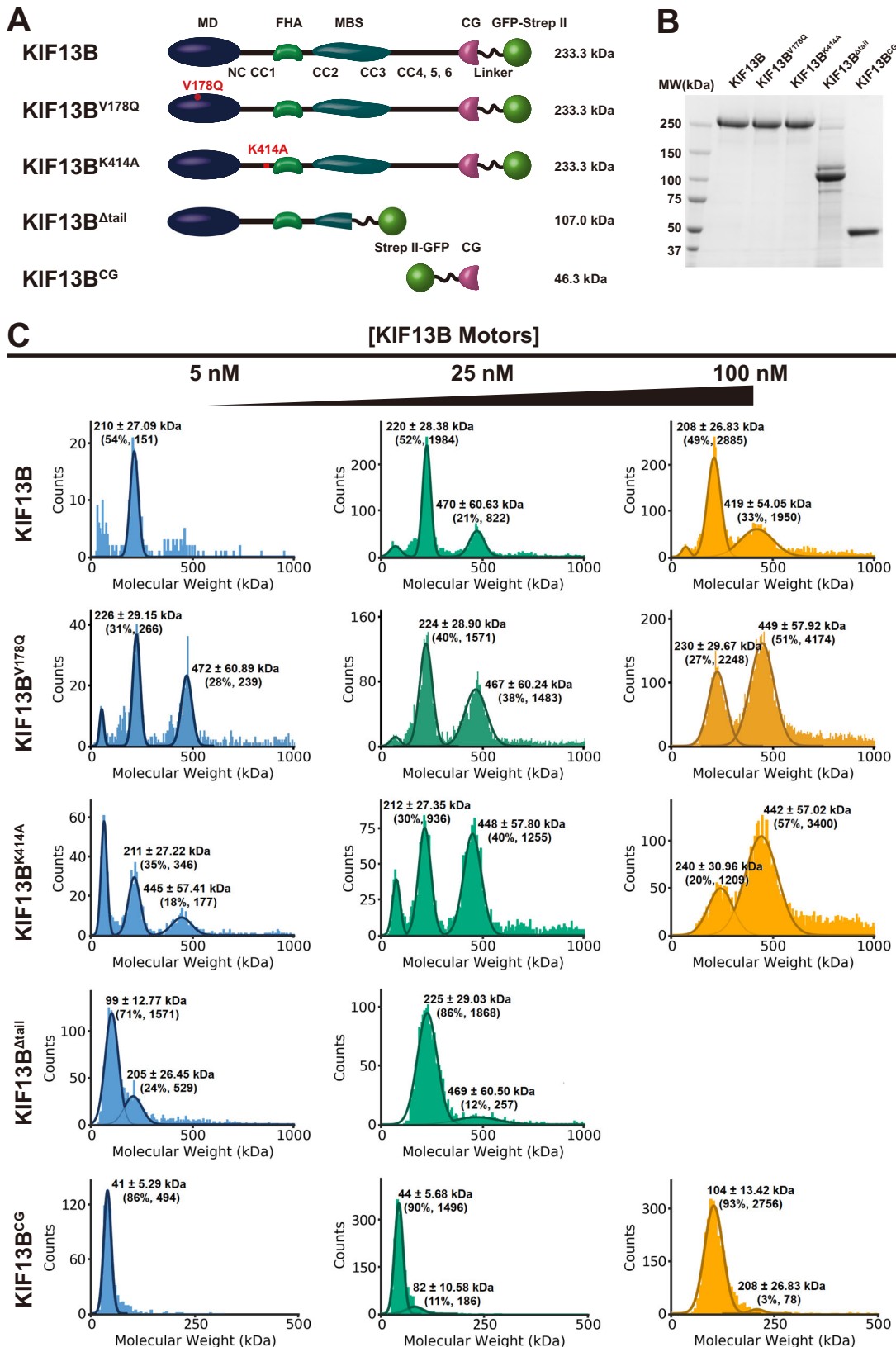

**Fig. 1 | Mass photometry analysis reveals the oligomeric states of recombinant KIF13B motors. A** Schematic of the KIF13B constructs used in this study. MD motor domain, FHA forkhead-associated domain, MBS membrane-associated guanylate kinase (MAGUK) binding stalk, CC coiled-coil, CG cytoskeleton-associated protein glycine-rich domain. Theoretical molecular weights of KIF13B motors are shown to the right. **B** SDS–PAGE analysis of purified KIF13B motors. **C** Characterization of the oligomeric state of recombinant KIF13B motors by mass photometry. Histograms in each panel show the particle counts of recombinant KIF13B motors at the indicated molecular mass. For each protein, the molecular mass was measured at two or three different indicated concentrations. The dark lines are Gaussian fits to the peaks. The calculated mass, percentage of particles in the peak, and the count of particles are indicated above each peak. KIF13B: $n = 315$, 3839, and 6010 molecules (at 5, 25, and 100 nM respectively), KIF13B[V178Q]: $n = 882$, 3948, and 8282 molecules. KIF13B[K414A]: $n = 1320$, 3337, and 6385 molecules. KIF13B[Δtail]: $n = 2537$ and 3895 molecules. KIF13B[CG]: $n = 623$, 1759, and 3622 molecules.

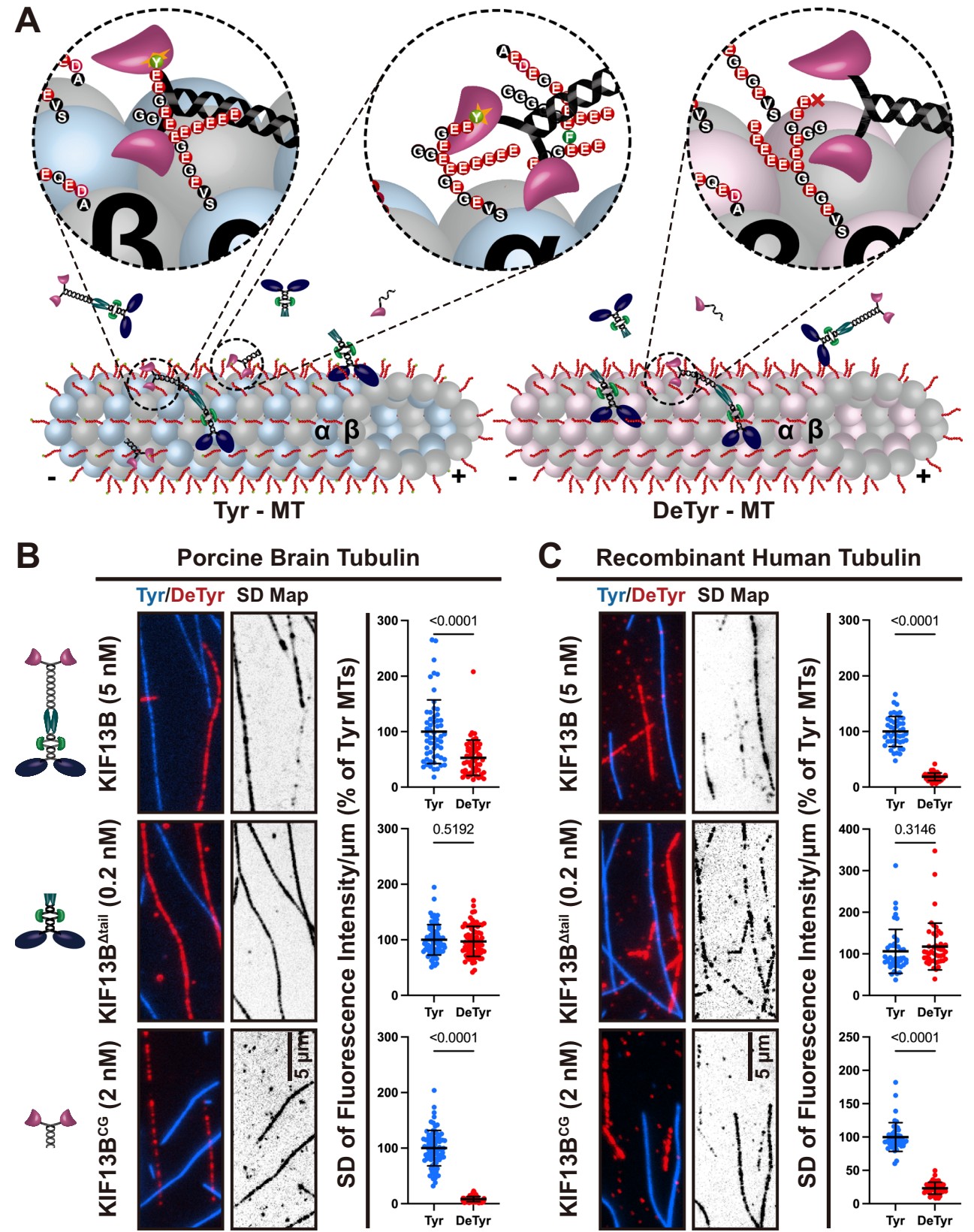

CPA-treated (detyrosinated) microtubules were mixed together in the same imaging chamber (Fig. 2B), providing an equal opportunity for motors to interact with each type of lattice.

Fluorescent proteins interacting transiently with the microtubule surface generate large pixel intensity variations over the background which can be averaged over an entire time series to generate a pixel-level fluorescence standard deviation (SD) map of the entire image series[29,55]. The SD map for full-length KIF13B revealed a robust preference for binding to tyrosinated versus detyrosinated microtubules in the presence of ATP (Fig. 2B). In contrast, KIF13B$^{\Delta tail}$, which lacks the CAP-Gly domain, bound equally to both types of microtubule (Fig. 2B), revealing that the KIF13B motor domain displays no preference for the

**Fig. 2 | KIF13B prefers to bind to tyrosinated microtubule. A** Schematic showing the interaction between CAP-Gly domain of KIF13B with the tyrosinated tail of α-tubulin within microtubules. Left: On tyrosinated microtubules, the CAP-Gly domain of KIF13B and KIF13B$^{\Delta motor}$ can interact with the tyrosinated tail of α-tubulin. Right: On detyrosinated microtubules, this interaction is abolished. Circled regions showing high magnification views of the interactions between CAP-Gly domain of KIF13B motors with tyrosinated tail of α-tubulin. Tyrosinated and detyrosinated α-tubulin are colored in light blue and light pink, respectively. **B** and **C** TIRF images of KIF13B motors bound to microtubules either polymerized from purified pig brain tubulin (**B**) or from recombinant human tubulin (**C**). Left panel, motors bound to tyrosinated (blue) or detyrosinated microtubules (red) in the presence of 2 mM ATP. Middle panel, SD map: standard deviation maps from an entire-time sequence reveal ensemble binding and dissociation events that lead to variations in pixel intensity, highlighting differences in binding between tyrosinated and detyrosinated microtubules. Scale bar: 5 μm. Right panel, quantification of mean intensity (arbitrary units) from the SD maps per μm microtubules for recombinant KIF13B motors bound to the tyrosinated or detyrosinated microtubules relative to tyrosinated microtubules in the same chamber. Microtubule polymerized from purified pig brain tubulin, KIF13B: $n = 57$, KIF13B$^{\Delta tail}$: $n = 80$, KIF13B$^{CG}$: $n = 80$. Microtubule polymerized from recombinant human tubulin, all KIF13B mutants, $n = 45$. Microtubules were quantified for each condition from two independent experiments. Mean ± SD is shown. *P* values are calculated from an unpaired, two-tailed *t*-test.

tyrosination state of the lattice. We further observed a strong enhancement of binding events on tyrosinated versus detyrosinated microtubules for KIF13$^{CG}$, confirming this construct binds microtubules predominantly via the CAP-Gly domain's interaction with the tyrosinated α-tubulin tail domain (Fig. 2B). Thus, we conclude that KIF13B's CAP-Gly domain plays a major role in the motor–microtubule interaction by strongly biasing the motor towards tyrosinated versus detyrosinated microtubules. Because the KIF13B motor domain is not affected by the tyrosination state of the lattice (Fig. 2B), these results reveal that the CAP-Gly domain enhances the initial microtubule encounter rate, interaction time of the motor on tyrosinated microtubules, or both. Because brain tubulin contains a mixture of tubulin isotypes and PTMs, we confirmed these observations with recombinant human tubulin composed of α1A /βIII tubulin isotypes containing either fully tyrosinated or detyrosinated alpha-tubulin[56]. All KIF13B constructs behaved similarly on recombinant tubulin (Fig. 2C). We conclude that differential KIF13B binding is due to the tyrosinated state of the lattice, and not from differences in tubulin isotypes or other PTMs.

## The CAP-Gly domain regulates the KIF13B's landing rate and run-length

We next sought to detail how the CAP-Gly domain of KIF13B affects the motile parameters of the motor during the initiation and continuation of processive movement along microtubules. We utilized single-molecule microscopy to track the behavior of full-length KIF13B, its two active mutants (KIF13B$^{V178Q}$ and KIF13B$^{K414A}$)[48], along with the tail-truncated KIF13B$^{\Delta tail}$ (Fig. 1A). Purified motors were introduced into flow chambers containing differentially labeled tyrosinated and detyrosinated microtubules at low enough concentrations to discern individual motors and their behavior over time in the presence of ATP. We observed processive motility along microtubules from all of the motor constructs examined (Fig. 3A). The landing rate of KIF13B was more than 50-fold less than that of truncated or mutant motors (Fig. 3B), reflecting the strong autoinhibition of the wild-type motor[48]. Experiments with dual-labeled motors, along with fluorescence brightness analysis further revealed that processive KIF13B$^{K414A}$ motors were dimeric, as expected (Supplementary Fig. 2). We noticed that all full-length KIF13B motors showed fewer processive events on detyrosinated versus tyrosinated microtubules, while the motility of KIF13B$^{\Delta tail}$ was largely similar between the two types of lattice (Fig. 3A, C). Indeed, quantification of the landing rates (all observable interactions of motors with the lattice) of each motor on each type of microtubule lattice revealed distinct differences between full-length motors containing the CAP-Gly domain, and KIF13B$^{\Delta tail}$. Full-length motors had approximately two-fold higher landing rates on tyrosinated versus detyrosinated microtubules (Fig. 3B), consistent with our fluorescence intensity analysis (Fig. 2B, C). In contrast, we observed no change in landing rates between the different types of microtubules for KIF13B$^{\Delta tail}$, revealing the preference for tyrosinated microtubules is driven by the tail domain of KIF13B. Importantly, we observed very similar results with detyrosinated microtubules

generated in vitro via CPA treatment, or treatment with the recently identified cellular tubulin detyrosinating enzyme complex, VASH1–SVBP[11,12] (Supplementary Fig. 3), confirming the specificity of CPA treatment.

We quantified KIF13B's motile parameters on tyrosinated and detyrosinated microtubules located in the same chamber. All KIF13B constructs moved along both types of microtubules with a similar velocity of ~1.25 μm/s, consistent with a previous study[49], revealing that the rate of the mechanochemical cycle of KIF13B is not affected by the tyrosination state of the microtubule (Fig. 3D, Supplementary Fig. 4A). We next measured motor run lengths on each type of microtubule. Full-length KIF13B motors showed long continuous runs (Fig. 3A, Supplementary Fig. 3B) with a fitted run length of ~5–6 μm on tyrosinated microtubules (Fig. 3E, Supplementary Fig. 4B). Strikingly, motor run-lengths were reduced approximately two to three-fold on detyrosinated microtubules for all full-length motors examined (Fig. 3E, Supplementary Fig. 4B), suggesting that an interaction of the tail domain with the tyrosinated microtubule lattice directly impacts motor processivity. In contrast, KIF13B$^{\Delta tail}$ had approximately two to three-fold shorter runs than full-length motors (Fig. 3E), further revealing that the rest of the tail domain enhances the processive movement of the motor, which we suggest may be through an interaction of the CAP-Gly domain with the tyrosinated microtubule lattice.

Importantly, no measurable difference in run-length was observed for KIF13B$^{\Delta tail}$ on the two types of microtubule lattice (Fig. 3E, Supplementary Fig. 4B). We again found very similar effects on the motile parameters of full-length motors on detyrosinated brain microtubules generated by either CPA or VASH1-SVBP, confirming the specificity of CPA treatment (Supplementary Fig. 4). Based on these data, we conclude that the CAP-Gly domain of KIF13B enhances both the motor's landing rate and processive run lengths, on tyrosinated microtubule lattices. In the absence of the CAP-Gly domain, the motor domain of KIF13B behaves indistinguishably on tyrosinated and detyrosinated microtubule lattices. Therefore, the tail domain of KIF13B, not the motor domain, recognizes the tyrosination state of the lattice and directly impacts the motility of the motor.

These results raised the possibility that the C-terminal CAP-Gly domain is continually engaged with the microtubule lattice during processive motility. Indeed, CAP-Gly domain-containing proteins, such as p150, have been observed to diffuse, along the microtubule lattice[57], providing a possible mechanism for interaction with the lattice during motility. To investigate this idea, we generated single chimeric microtubules composed of differentially labeled tyrosinated and detyrosinated sections of lattice[29] (Fig. 3F). Using this system, we could observe how individual KIF13B motors behaved as they traversed from one type of lattice onto the other. The majority of motors did not show obvious changes in the motile behavior as they traversed from one type of lattice to the other, but instead continued processive motility unabated (Fig. 3F). We quantified the fraction of motors that smoothly traversed the junction between each type of lattice and found that ~90% continued their processive motility while changing onto a different type of lattice (Fig. 3G). Because the CAP-Gly domain binds very

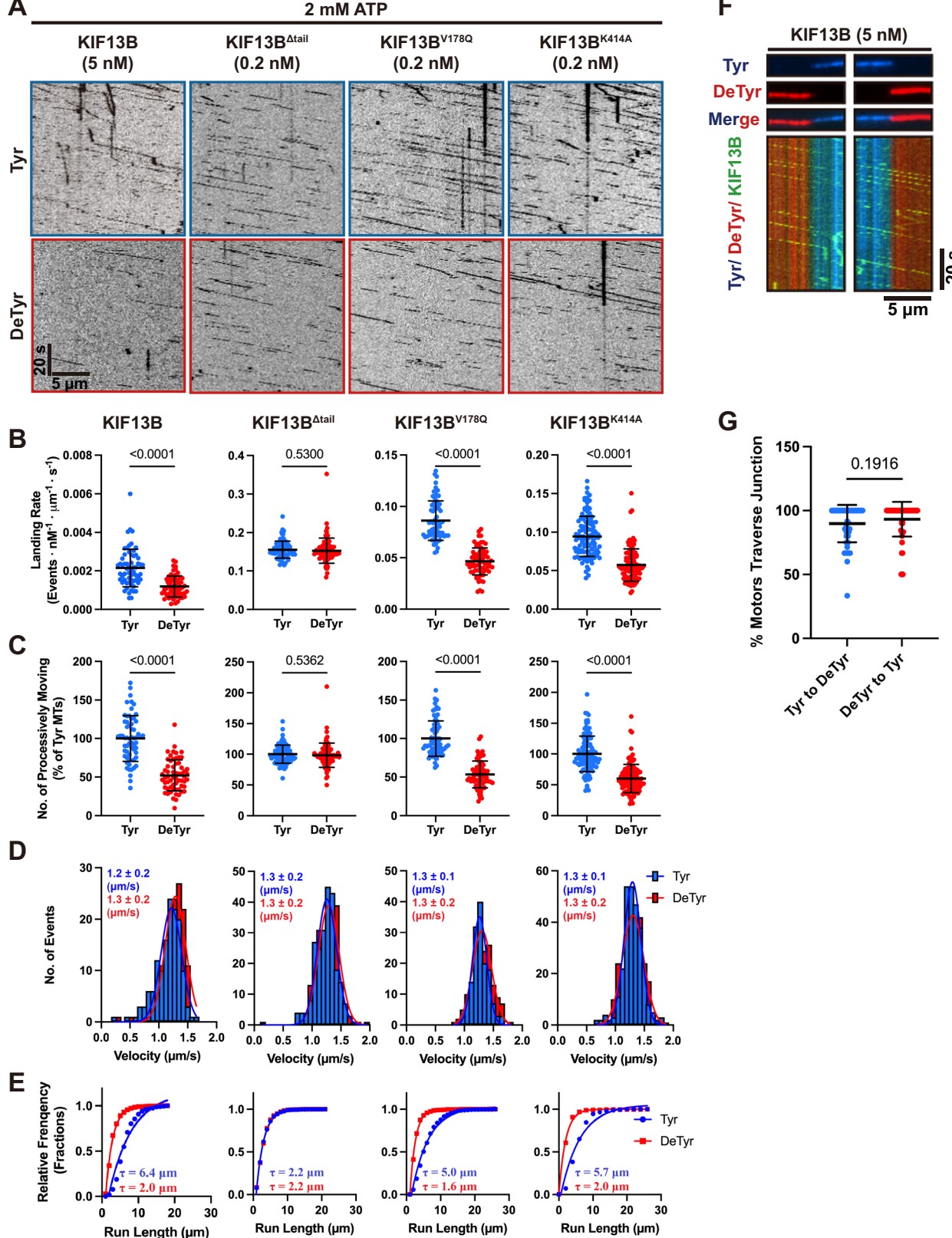

weakly to detyrosinated lattices (Fig. 2B), we conclude that processive KIF13B motility does not require continuous CAP-Gly engagement with the microtubule lattice. However, the strong effect of detyrosination on motor run lengths (Fig. 3E) leads us to suggest that the CAP-Gly domain may allow for rapid motor re-attachment to the tyrosinated microtubule lattice upon dissociation of the motor domains during a processive run, facilitating the continuation of processive motility.

## The motor's nucleotide state influences its microtubule preference

Having observed that the KIF13B CAP-Gly domain regulates the motor's landing rate and processive motility, we next sought to investigate the molecular mechanism behind these observations. KIF13B contains two distinct types of microtubule-binding domains. The kinesin motor domain cycles between a weak microtubule binding

**Fig. 3 | Regulation of motility by the CAP-Gly Domain. A** Kymographs of KIF13B motors on tyrosinated or detyrosinated microtubules in 2 mM ATP. Scale bars: 20 s and 5 μm. **B** Quantified landing rates of KIF13B motors. **C** Quantification of the number of processive motors from microtubules within the same chamber. For **B** and **C**, microtubules were quantified from two independent experiments. KIF13B: $n = 61$ (tyrosinated and detyrosinated microtubules). KIF13B$^{\Delta tail}$: $n = 81$. KIF13B$^{V178Q}$: $n = 72$. KIF13B$^{K414A}$: $n = 110$). Mean ± SD is shown. $P$ values were calculated from an unpaired, two-tailed $t$-test. **D** Velocities of KIF13B motors on tyrosinated and detyrosinated microtubules. Histograms of the velocities and fits to a single Gaussian distribution are shown. Events were quantified for each condition from two independent experiments. KIF13B: $n = 140, 152$ (on tyrosinated or detyrosinated microtubules). KIF13B$^{\Delta tail}$: $n = 232, 234$. KIF13B$^{V178Q}$: $n = 152, 169$. KIF13B$^{K414A}$: $n = 281, 264$. The mean ± SD is indicated in the top left corners. **E** Cumulative frequency of the run lengths is plotted for each population of motors and fit to a one-phase exponential decay function. Events were quantified for each condition from two independent experiments. The fitted run lengths ($\tau$) are indicated. KIF13B: $n = 414, 583$ and $R^2 = 0.958, 0.995$ (on tyrosinated and detyrosinated microtubules). KIF13B$^{\Delta tail}$: $n = 1004, 933$ and $R^2 = 0.999, 0.997$. KIF13B$^{V178Q}$: $n = 517, 488$ and $R^2 = 0.985, 0.997$. KIF13B$^{K414A}$: $n = 506, 528$ and $R^2 = 0.958, 0.997$. **F** TIRF images (top) and kymographs (bottom) from chimeric microtubules composed of tyrosinated and detyrosinated microtubules. Motors moving either from detyrosinated onto tyrosinated microtubules (left panel in the bottom) or from tyrosinated onto detyrosinated microtubules (right panel in the bottom) are shown. Scale bars: 20 s and 5 μm. **G** Quantification of the number of processive motors of KIF13B that traverse the microtubule boundaries and continue moving unimpeded. Motors were quantified for each condition from two independent experiments, $n = 66, 54$ for motors moving from tyrosinated to detyrosinated microtubules and from detyrosinated to tyrosinated microtubules respectively. Mean ± SD is shown. $P$ value was calculated from an unpaired, two-tailed $t$-test.

state (ADP state) and a strong microtubule binding state (ATP and apo states)[58]. In contrast, the non-enzymatic CAP-Gly domain only depends on the tyrosination state of the lattice for its interaction with microtubules. We sought to understand the relative contributions of each domain within the context of the KIF13B mechanochemical cycle. We performed microtubule binding assays with tyrosinated and detyrosinated microtubules in the same chamber, and KIF13B motors in either saturating ADP to lock the motor domain in the weak binding state, or the ATP analog adenylyl-imidodiphosphate (AMP-PNP) to lock the motor domain in the strong binding state (Fig. 4A). In the presence of ADP, we observed that full-length KIF13B binding to tyrosinated microtubules was ~8-fold higher than on detyrosinated microtubules in the same chamber (Fig. 4B), revealing that when the motor domain is bound to ADP, KIF13B's interaction with the microtubule is strongly influenced by the CAP-Gly domain. In contrast, KIF13B binding to tyrosinated microtubules was only ~1.4-fold higher than to detyrosinated microtubules in AMP-PNP (Fig. 4B), suggesting that in the strong binding state of the ATPase cycle, the affinity of the motor domain for the lattice blunts the preference of the CAP-Gly domain for tyrosinated microtubules. The tail-truncated construct, KIF13B$^{\Delta tail}$, showed no difference in binding to tyrosinated or detyrosinated microtubules in the presence of nucleotide (Fig. 4B), consistent with our prior results that the motor domain of KIF13B is not affected by the tyrosination state of the lattice (Fig. 3A, B).

To understand how autoinhibition and dimerization affect the ability of the CAP-Gly domain to bias KIF13B to tyrosinated microtubules, we directly compared the fluorescence intensity of KIF13B and KIF13B$^{K414A}$ in the presence of ADP. We assayed the motors at 50 nM, a concentration at which we expected the fraction of dimeric KIF13B$^{K414A}$ motors to be substantially higher than that for KIF13B (Fig. 1C). Comparison of the average motor density along microtubules revealed substantially higher levels of KIF13B$^{K414A}$ bound to both tyrosinated and detyrosinated microtubules as compared to KIF13B (~3-fold and 11-fold higher for KIF13B$^{K414A}$ on tyrosinated and detyrosinated microtubules, respectively, Fig. 4C). Importantly, an ~2-fold preference of KIF13B$^{K414A}$ for tyrosinated over detyrosinated microtubules remained, consistent with the idea that the CAP-Gly domain directs the motor's interaction with microtubules when the motor domain is bound to ADP. Titration of the motor concentration across a 500-fold concentration range (0.1–50 nM) revealed an apparent $K_d$ of 23.7 and 72.7 nM ($n = 160$ MTs measured, $N = 2$) for tyrosinated and detyrosinated microtubules, respectively. We conclude that release from autoinhibition and dimerization of the motor results in a higher affinity for both types of microtubules, but the CAP-Gly domain provides an approximately 3-fold higher affinity for tyrosinated microtubules when the motor domain is locked in the ADP state (Fig. 4C).

Because the amount of KIF13B$^{K414A}$ bound to detyrosinated microtubules in the presence of ADP is much higher than KIF13B, we

hypothesize that that avidity arising from dimerization of the motor enhances the binding affinity of both the CAP-Gly and motor domains within KIF13B. In support of this idea, we observed dual-labeled KIF13B$^{K414A}$ motors transiently binding to the microtubule in ADP and the measured GFP fluorescence intensity of both KIF13B$^{K414A}$ and KIF13B$^{CG}$ molecules bound to microtubules was largely consistent with the brightness of a dimeric kinesin-1 (KIF5B) motor (Supplementary Fig. 2A–C), suggesting that binding to microtubules induces dimerization of the isolated CAP-Gly domain. In contrast, KIF13B$^{CG}$ which was not bound to microtubules in the same chamber was dimmer on average, suggesting a more monomeric population, as expected from our mass photometry data (Fig. 1, Supplementary Fig. 2B–D). In addition to potential avidity effects, the increased affinity is also likely due to the microtubule-stimulated release of ADP from the motor domain, which is enhanced upon release from the autoinhibited state[48]. From these data, we conclude that KIF13B's preference for tyrosinated microtubules is strongly influenced by the nucleotide state of the motor domain. In the ADP state, the CAP-Gly domain dominates the interaction between KIF13B and microtubules, resulting in a prominent preference for tyrosinated microtubules. In the ATP state (mimicked by AMP-PNP), the motor domain dominates the interaction between KIF13B and microtubule, and the preference for tyrosinated microtubules is diminished. Because ADP release is the rate-limiting step in the kinesin mechanochemical cycle and is greatly accelerated by microtubule binding[58], the motor domain is expected to accumulate in this state prior to association with the microtubule, suggesting that the CAP-Gly domain will substantially influence the initial encounter of KIF13B with microtubules in cells.

To explore this point further, we created chimeric construct consisting of the constitutively activated N-terminal motor domain and stalk region of KIF5B, with its autoregulatory tail region removed (KIF5B$^{912}$) or appended with the CAP-Gly domain from KIF13B (KIF5B$^{CG}$, Supplementary Fig. 5A). The wild-type kinesin-1 construct moved processively along both tyrosinated and detyrosinated microtubules, where it showed a small preference for interaction with detyrosinated microtubules (Supplementary Fig. 5B–D), consistent with prior findings[59]. In contrast, KIF5B$^{CG}$ displayed a large preference for landing on tyrosinated microtubules, suggesting the KIF13B CAP-Gly domain appended to its tail domain strongly biased the motor's initial encounter with the microtubule (Supplementary Fig. 5B–D). While the tyrosination state of the lattice did not impact the motor velocity for either construct (Supplementary Fig. 5E), we observed enhanced run-lengths of the chimeric construct along tyrosinated microtubules (Supplementary Fig. 5F), consistent with our observations for KIF13B (Fig. 3E). In summary, these results support our findings for KIF13B and reveal that an exogenous CAP-Gly domain placed within an orthogonal kinesin tail domain strongly influences the motor's interaction with tyrosinated microtubules, in congruence with of our findings for native KIF13B (Fig. 3).

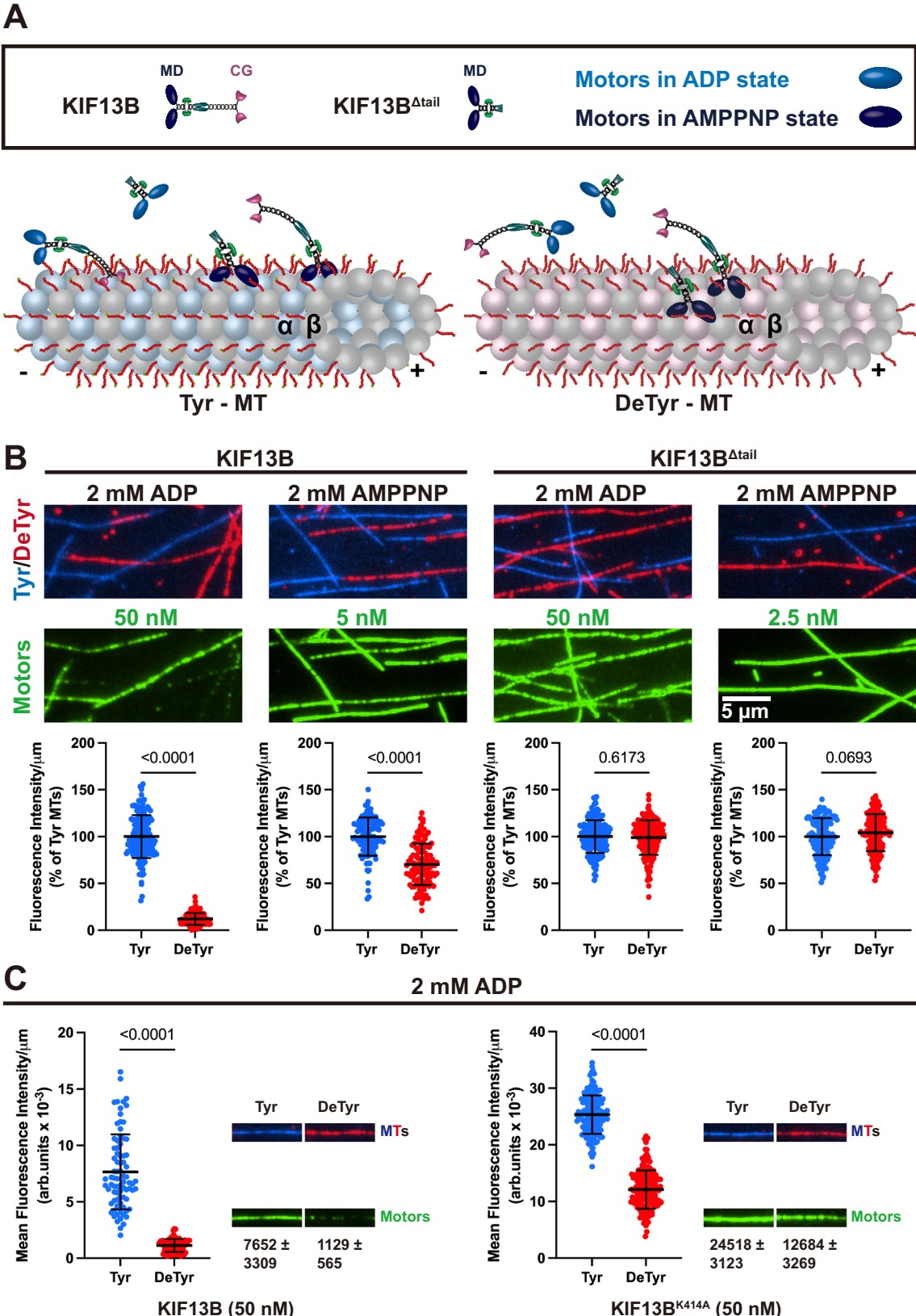

## KIF13B's CAP-Gly enhances its microtubule preference in the ADP state

To further understand how the CAP-Gly domain affects KIF13B's interaction with microtubules, we analyzed the microtubule-binding behavior of KIF13B and its mutants on tyrosinated and detyrosinated microtubules in the presence of ADP. Kymographs showed that single

molecules of KIF13B, KI13B[K414A], and KI13B[ΔTail] bound to both types of microtubules. For KI13B[CG], only very few binding events could be detected on detyrosinated microtubules as expected (Fig. 5A). Quantification of the landing rates revealed that both KIF13B and KIF13B[K414A] showed a strong ~2-fold preference for tyrosinated microtubules (Fig. 5B). Consistent with the idea that this preference is driven by the

**Fig. 4 | Nucleotide-dependent preference for tyrosinated microtubules of KIF13B. A** Schematic showing interactions between KIF13B and tyrosinated or detyrosinated microtubules in the presence of 2 mM ADP or 2 mM AMPPNP. Left: motors binding along the tyrosinated microtubules. In the ADP state (motor domains: light blue), the CAP-Gly domain dominates the interaction with microtubules. In the AMPPNP state (motors domains: dark blue), the motor domain dominates the interaction with microtubules. Right: motors binding along the detyrosinated microtubules. Since the interaction between CAP-Gly and α-tubulin is abolished, the motor domain dictates the interaction between microtubules and motors in either ADP or AMPPNP states. MD motor domain, CG CAP-Gly domain. **B** The fluorescence intensity of KIF13B and KIF13B$^{\Delta tail}$ on tyrosinated and detyrosinated microtubules in different nucleotide states. TIRF images of KIF13B and KIF13B$^{\Delta tail}$ (green, middle panel) bound to either tyrosinated (blue, top panel) or detyrosinated microtubules (red, top panel). Bottom panel, quantification of mean fluorescence intensity (arbitrary units) per μm microtubules for motors bound to tyrosinated or detyrosinated microtubules relative to tyrosinated microtubules in

the same chamber. Microtubules were quantified for each condition from two independent experiments. KIF13B (in AMPPNP state): $n = 116$ (both tyrosinated and detyrosinated microtubules). KIF13B$^{\Delta tail}$ (in AMPPNP state): $n = 150$. KIF13B (in ADP state): $n = 164$. KIF13B$^{\Delta tail}$ (in ADP state): $n = 160$. Scale bar: 5 μm. Mean ± SD is shown. $P$ values are calculated from an unpaired, two-tailed $t$-test. **C** Comparison of the microtubule-binding of KIF13B motors on tyrosinated and detyrosinated microtubules in the presence of 2 mM ADP. 50 nM of KIF13B or KIF13B$^{K414A}$ was incubated with tyrosinated and detyrosinated microtubules in chambers for 10 min, and mean fluorescence intensities of the MT-bound KIF13B (left) or KIF13B$^{K414A}$ (right) were quantified (left side bar graphs). Representative microtubule images and the corresponding motor channels are shown on the right side in each condition. The width of each panel corresponds to 8.06 μm. The mean fluorescence intensity of motors ± SD is shown below. For quantification, microtubules were quantified for each condition from two independent experiments. For each condition, KIF13B: $n = 80$. KIF13B$^{K414A}$: $n = 160$. $P$ values are calculated from an unpaired, two-tailed $t$-test.

CAP-Gly domain. KIF13B$^{CG}$ displayed a much larger ~14-fold preference for tyrosinated microtubules, similar to our fluorescence intensity analysis (Fig. 2B, C). In contrast, KIF13B$^{\Delta tail}$ again showed no preference for either type of microtubule (Fig. 5B). However, its landing rate was an order of magnitude higher than full-length motors or KIF13B$^{CG}$, suggesting the truncation of the tail releases the motors from auto-inhibition. Two-color analysis revealed that the majority of KIF13B$^{K414A}$ molecules that bound to microtubules in ADP were dimeric (Supplementary Fig. 2A), despite the low concentrations used in this assay.

We next measured the dwell times of motors from the resulting kymographs. This analysis revealed that while KIF13B$^{\Delta tail}$ showed highly similar dwell times on both types of microtubules, the other constructs tested showed longer dwell time events on tyrosinated microtubules (Fig. 5A–C). We found that the dwell times of KIF13B and KIF13B$^{K414A}$ on tyrosinated microtubules were ~1.3 and ~1.5 times higher than on detyrosinated microtubules, respectively. For KIF13B$^{CG}$, the dwell time on tyrosinated microtubules is ~2–3 times lower than for full-length KIF13B or KIF13B$^{\Delta tail}$, but the landing preference for tyrosinated microtubules was the strongest of the constructs tested. Consistently, KIF13B$^{\Delta tail}$ showed no difference in dwell times (Fig. 5C). These results reveal that the CAP-Gly domain enhances the motor's dwell time on tyrosinated microtubules when the motor domain is in the ADP state, revealing a possible mechanism for CAP-Gly dependent initiation of processive motility on tyrosinated microtubules. The longer dwell times for full-length motors compared to KIF13B$^{CG}$ could reflect a contribution of the ADP-bound motor domain to the KIF13B-microtubule interaction. The landing rates of the full-length KIF13B motor are an order of magnitude lower than that of KIF13B$^{\Delta tail}$ and KIF13B$^{K414A}$, and similar to KIF13B$^{CG}$, suggesting the motor domains in the wild-type motor are more strongly inhibited from binding to microtubules, resulting in the predominance of the CAP-Gly domain interaction with the tyrosinated microtubule. However, the dwell times of full-length KIF13B are longer than KIF13B$^{CG}$, suggesting a contribution of the ADP-motor domains once the motor is engaged with the microtubule.

To further corroborate these findings, we examined the binding of KIF5B$^{912}$ or KIF5B$^{CG}$ to microtubules in the presence of ADP. KIF5B$^{912}$ showed a slight preference for detyrosinated microtubules as noted above, while the KIF5B$^{CG}$ chimeric motor showed a large ~16-fold preference for binding tyrosinated over detyrosinated microtubules in ADP (Supplementary Fig. 5G). These results further confirm that a CAP-Gly domain located at the extreme C-terminus of a kinesin molecule strongly influences the microtubule preference of the motor when the motor domain is bound to ADP.

### Model for the role of the CAP-Gly domain in KIF13B motility

Our data reveal at least two roles of the CAP-Gly domain in KIF13B motility. Firstly, when the motor domains are bound to ADP and not

strongly engaged with the microtubule lattice, the CAP-Gly domain biases the interaction of the motor with tyrosinated microtubules via its binding of the α-tubulin C-terminal tyrosine residue (Fig. 6, step 1). The oligomerization state of the motor at this step is currently unclear (Fig. 6, question mark), but we suggest that the binding of the CAP-Gly domain to the lattice could facilitate localized concentration of motors on the lattice, in turn stimulating dimerization and activation of processive motility. Indeed, the isolated CAP-Gly domain appears to dimerize when bound to microtubules (Supplementary Fig. 2C), which may also suggest a role for this domain in facilitating motor dimerization, but further work is needed to clarify this point. It is also conceivable that engagement of the CAP-Gly domain with the microtubule leads to molecular rearrangements of the motor from the auto-inhibited, monomeric state, priming the motor for dimerization on the microtubule surface. Although we could not detect such events in our data (Supplementary Fig. 2A), it is possible the dimerization event happens too quickly to observe within the time resolution of our experiments. A similar mechanism has recently been suggested for myosin-7a[60]. Alternatively, binding of cargo to the motor may stimulate dimerization prior to microtubule association[49]. Association of the CAP-Gly with the microtubule could facilitate the binding of the motor domains to the lattice, leading to nucleotide exchange (Fig. 6, step 2), and subsequent processive motility (Fig. 6, step 3). During processive motility, the motor domains remain out of phase with respect to the nucleotide state such that one motor domain is always tightly bound to the lattice in the ATP state. However, at some frequency, both motor domains enter the weak binding ADP state simultaneously, leading to motor dissociation from the lattice. When both motor domains are in the ADP state, our data reveals that the CAP-Gly domain binding to tyrosinated microtubules dominates the overall microtubule-motor interaction (Fig. 4B). Because of this, the CAP-Gly domain may facilitate fast rebinding of the motor to the lattice, preventing diffusion away from the microtubule and termination of motility (Fig. 6, step 4). In support of this idea, we found that insertion of the KIF13B CAP-Gly domain onto the C-terminus of KIF5B leads to enhanced run lengths on tyrosinated microtubules (Supplementary Fig. 5F).

### Discussion

Kinsein-3 motors transport cargo over long distances within cells and are critical for human health and disease[61]. As such, understanding the molecular mechanisms underpinning the functions of this kinesin family represents an important question in cell biology. Here we have studied the function of the unique tail domain of KIF13B, which contains a specialized second microtubule binding site in its CAP-Gly domain. Our work revealed that KIF13B's CAP-Gly domain drives a strong preference for motor binding to tyrosinated MTs in vitro and that this preference depends on the nucleotide state of the motor

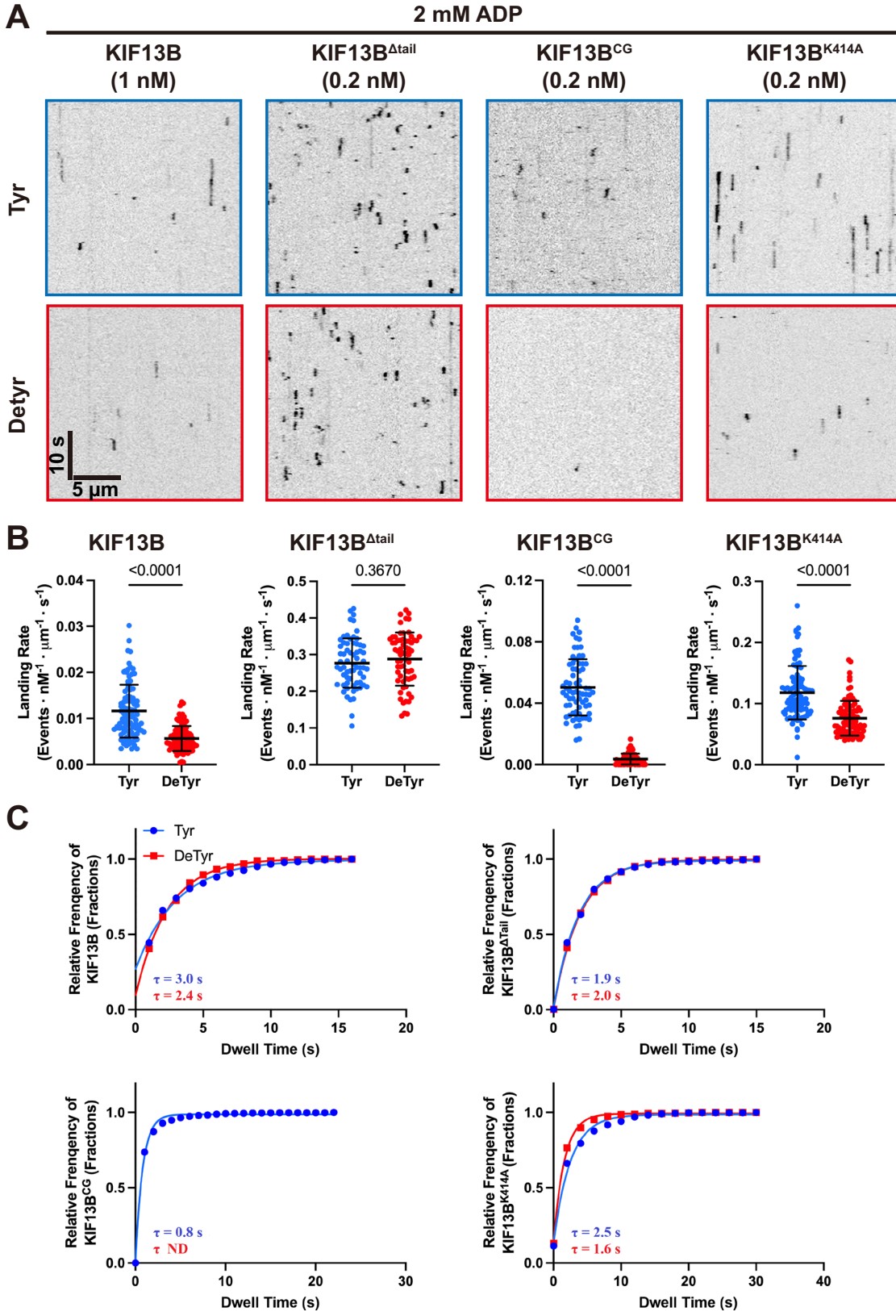

domain. In addition, the CAP-Gly domain mediates higher landing rates and an approximately 3-fold longer run length on tyrosinated versus detyrosinated MTs. Given that defects in kinesin-3's interactions with microtubules underly human neurodevelopmental and neurodegenerative diseases[62,63], our results provide insights into the various strategies adopted by kinesin-3 motors to augment the ability of the motor

domains to move processively along microtubules to support cellular trafficking homeostasis and human health.

Previous studies have revealed that the CAP-Gly domain-containing proteins CLIP-170 and CLIP-115 are involved in the regulation of microtubule dynamics, via their specific recruitment to the dynamic plus end of MTs[25,26]. The CAP-Gly domains of these proteins directly

**Fig. 5 | CAP-Gly domain enhances the interaction between KIF13B and tyrosinated microtubule in the ADP state. A** Kymographs showing KIF13B motors binding to tyrosinated (top panel) and detyrosinated microtubules (bottom panel) in the presence of 2 mM ADP. Scale bars: 10 s and 5 μm. **B** Quantification of landing rates of KIF13B motors on tyrosinated and detyrosinated microtubules in ADP state. For quantification, microtubules were quantified for each condition from two independent experiments. KIF13B: $n = 96$ (both on tyrosinated and detyrosinated microtubules). KIF13B$^{\Delta tail}$: $n = 65$. KIF13B$^{CG}$: $n = 70$. KIF13B$^{K414A}$: $n = 94$). Mean ± SD is shown. $P$ values are calculated from an unpaired, two-tailed $t$-test. **C** Quantification of the dwell times of KIF13B and its mutants on tyrosinated or detyrosinated microtubules. Cumulative frequency of the dwell time is plotted for each population of motors and fit to a one-phase exponential decay function. The characteristic dwell times derived from the fits ($\tau$) are indicated in the bottom left corners. Events longer than three pixels were selected and quantified and two independent experiments were performed for each condition. KIF13B: $n = 414, 317$ and $R^2 = 0.989, 0.999$ (on tyrosinated and detyrosinated microtubules, respectively). KIF13B$^{\Delta tail}$: $n = 433, 470$ and $R^2 = 0.999, 0.9998$. KIF13B$^{CG}$: $n = 1635$, $R^2 = 0.991$. KIF13B$^{K414A}$: $n = 520, 482$ and $R^2 = 0.989, 0.997$. ND no data.

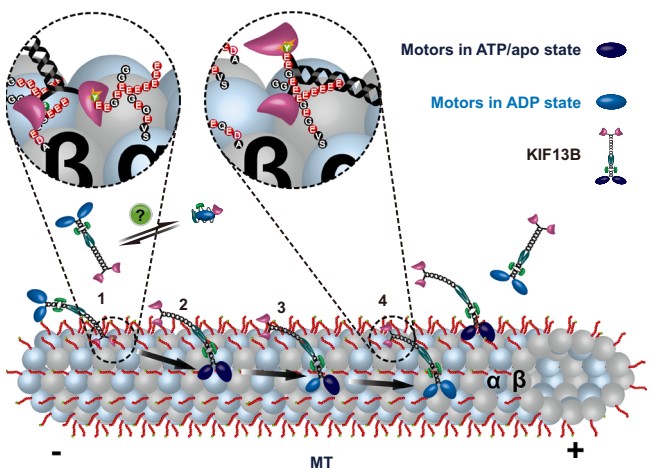

**Fig. 6 | Model for the regulation of CAP-Gly domain in KIF13B motility.** Schematic showing a model for the roles of the CAP-Gly domain in KIF13B motility: (1) In the ADP state, the CAP-Gly domain biases the interaction between KIF13B and microtubules. Interaction between the CAP-Gly domain and the tyrosinated α-tubulin within microtubules facilitates the binding of the motor domain to microtubule lattice. (2) Landing of the motor domain onto the microtubule stimulates ADP release, leading to nucleotide exchange, which in turn increases affinity between the KIF13B motor domain and microtubules. (3) Nucleotide exchange begins processive motility along microtubules powered by asynchronous ATP hydrolysis in the two motor domains. In this process, at least one motor domain remains strongly bound to the lattice until (4) at some time frequency, both motor domains simultaneously enter the weak binding ADP state, resulting in motor release from the lattice. In this case, the interaction between the CAP-Gly domain and the tyrosinated tail of α-tubulin facilitates the rebinding of the motor to the lattice and initiates a new mechanochemical cycle.

interact with the EEY/F motif of α-tubulin[20,21] and a similar motif found within end-binding (EB) proteins[25,64]. The microtubule plus end localization of these proteins stimulates the dynamicity of MTs by increasing catastrophe and rescue frequencies[25,26]. Given that many CAP-Gly proteins are recruited to dynamic microtubule plus-ends through interactions with EB proteins, we note that despite a current lack of evidence, it is formally possible that KIF13B is also recruited to microtubule plus-ends in a similar manner. Human KIF13B contains several candidate SxIP motifs known to be crucial for interactions with EBs[65], and future work should focus on a potential role for the plus-end recruitment of KIF13B, possibly to aid in the delivery of cortical machinery necessary for proper mitotic spindle orientation[66].

In the context of motor protein movement, in vitro reconstitution showed that the CAP-Gly domain within the p150 subunit of the dynactin complex strongly modulates the initiation of dynein-driven motility along tyrosinated MTs without affecting the continuous movement of the DDB complex[29], while the domain is also important for the initiation of retrograde movements in cells[30]. These studies illustrate that CAP-Gly domain-mediated interactions are utilized by cells to fine-tune both microtubule dynamics and the initiation of movement of motor proteins. We now extend these findings to the kinesin family of anterograde motors and suggest that KIF13B cargo transport dynamics within cells are likely influenced by the tyrosination state of the local microtubule lattice.

KIF13B has been reported to transport various cargos within cells, such as phosphatidylinositol 3,4,5-triphosphate (PIP3) containing lipid vesicles, low-density lipoprotein (LDL) receptor-related protein 1 (LRP1) containing vesicles, vascular endothelial growth factor receptor 2 (VEGFR2), rab6 vesicles and myosin X[35–45]. The molecular basis of these diverse motor–cargo interactions remains unclear, but some interactions involve the forkhead-associated (FHA) domain or MBS regions of the motor[42,45,51] (Fig. 1A). To date, a direct role for the distal CAP-Gly domain in cargo-binding has not been described, but its deletion in mice results in reduced uptake of LRP1 and ultimately resulted in higher serum cholesterol levels[38]. These results reveal a direct role of the CAP-Gly domain in receptor-mediated endocytosis and trafficking of lipoproteins[38]. Our results imply that the misdirection of the motor along the differentially tyrosinated microtubules may underlie this observation.

KIF13B is not the only kinesin that contains an auxiliary microtubule-binding domain. Kinesin-1 is also reported to contain a secondary microtubule-binding site composed of charged residues located within the C-terminal tail of the heavy chain, which is important for its ability to reorganize the cytoskeleton by sliding microtubules relative to one another[67,68]. The highly processive motor kinesin-8 (mammalian KIF18A and yeast Kip3p) contains a microtubule-binding site in its tail domain essential for correct plus-end localization of this motor in mitosis[69]. Therefore, specialized microtubule binding sites located distal to the kinesin motor domain are an apparently malleable evolutionary adaptation utilized to diversify kinesin functions in cells, and our results build on this repertoire within the kinesin family.

Finally, a recent structure of the autoinhibited kinesin-3 family member KLP-6 revealed a multimodal mechanism for keeping the monomeric motor in an inactive state[52]. The mechanism involves intricate folding of most of the kinesin tail domain around the motor domain, sterically occluding the microtubule binding site and trapping the motor in the ADP state. This mechanism likely has strong parallels in KIF13B, as the domain organization of these motors is highly similar. If the CAP-Gly domain of KIF13B plays a role in the autoinhibition of the motor is an interesting question for future structural studies of KIF13B and its homologs. Our in vitro reconstitution with full-length KIF13B will provide a platform to examine hypotheses about the mechanism and regulation of this important transport kinesin.

## Methods

### Plasmids
Full-length human KIF13B cDNA was purchased from Transomics (BC172411). A DNA fragment encoding the sfGFP-StrepII tag was synthesized by gBlocks (Integrated DNA Technologies, Coralville, IA, USA). To generate insect cell expression constructs, Coding sequences of human KIF13B, KIF13B$^{V178Q}$, KIF13B$^{K414A}$, and KIF13B$^{\Delta tail}$ (a.a. 1–679) were tagged with sfGFP-StrepII in C-terminal and cloned into pAceBac1 vector (Geneva Biotech) by Gibson Assembly. The coding sequences of human KIF13B$^{CG}$ (a.a. 1685–1826) were cloned into pET28a vector by

Gibson Assembly with a sfGFP-StrepII tag fused to its N-terminal via a 3xGGGGS linker. The coding sequence of full-length KIF13B, KIF13B$^{\Delta tail}$, KIF13B$^{CG}$ sfGFP-StrepII, and 3×GGGGS linker were amplified by PCR, all point mutations were introduced by PCR-based mutagenesis. Plasmids encoding recombinant tubulins (tubulin variants α1A/βIII and α1A-Y/βIII in pFAST$^{TM}$-Dual vector) were kind gifts from Antonina Roll-Mecak[61] (Cell Biology and Biophysics, National Institutes of Health). All constructs were verified by Sanger sequencing.

## Protein expression and purification

KIF13B, KIF13B$^{VI78Q}$, KIF13B$^{K414A}$, and KIF13B$^{\Delta tail}$ were expressed in insect cells. Briefly, insect Sf9 cells were grown or maintained in Sf-900 II serum-free medium (Thermo Fisher Scientific) or in ESF medium (Expression Systems) at 27 °C. Plasmids DNA of corresponding constructs were transformed into DH10EmBacY competent cells (Geneva Biotech) to generate bacmids. To get baculovirus, $1 \times 10^6$ Sf9 cells were transferred to each well of a six-wells plate and subsequently transfected with 1 µg of bacmid DNA by 6 µl of Cellfectin (Thermo Fisher Scientific). Baculovirus-containing cell supernatants (P1) were harvested 7 days (cells in Sf-900 II medium) or 11 days (cells in ESF medium) after transfection when the entire cell population has become infected. To amplify the baculovirus (P2), 50 ml of Sf9 cells were infected with 50 µl of P1 virus, and the P2 virus was harvested 4 days (cells in Sf-900 II medium) or 7 days (cells in ESF medium) after infection. To prepare recombinant proteins, 200 ml of Sf9 cells at high density (-2 × 10^6 cells/ml) were infected by P2 virus (virus:cells = 1:100, v/v) and cultured around 65 h at 27 °C. Cells were subsequently harvested by centrifuge at 1000 × g for 10 min and flash frozen by liquid nitrogen.

KIF13B$^{CG}$ was expressed in BL21-CodonPlus (DE3)-RIPL cells (Agilent). Briefly, a single colony was inoculated into 50 ml LB Broth medium (Fisher Scientific) and grew overnight at 37 °C, the culture was then diluted 1–500 into 500 mL LB Broth medium (Fisher Scientific) and cultured at 37 °C until the optical density at 600 nm (OD$_{600}$) reaches 0.4. Expression was induced 18 h by 0.5 mM isopropyl-β-d-thiogalactoside (Fisher Scientific) at 24 °C. Cells were subsequently harvested by centrifuge at 1000×g for 10 min and flash frozen by liquid nitrogen. All frozen cells were stored at −80 °C before purification.

To purify KIF13B, KIF13B$^{VI78Q}$, KIF13B$^{K414A}$, and KIF13B$^{\Delta tail}$, the frozen pellet was thawed on ice and resuspended in 40 ml of PB buffer (purification buffer: 50 mM Tris–HCl pH 8.0, 150 mM KCH$_3$COO, 2 mM MgSO$_4$, 1 mM EGTA, 5% glycerol) freshly supplemented with 1 mM dithiothreitol (DTT), 1 mM phenylmethylsulfonyl fluoride (PMSF), 1 mM ATP, and protease inhibitor mix (Promega). Resuspended cells were homogenized in a dounce homogenizer and lysed by the addition of 1% Triton X-100 and 5 µl of Benzonase Nuclease (Millipore) for 10 min on ice. To obtain soluble lysate, lysed cells were centrifuged at 15, 000×g for 20 min at 4 °C. For affinity purification, the clarified lysate was incubated with 2 ml of Strep-Tactin XT resin (IBA) for 1 h at 4 °C and washed with PB buffer, protein was subsequently eluted by elution buffer (PB buffer plus 100 mM biotin, pH 8.0). All proteins were further purified by gel filtration. Briefly, eluted protein was concentrated to 500 µl via Amicon Ultra 5 centrifugal filters (Merck) and separated by Phenomenex Yarra 3 µm SEC-4000 300 × 7.8 mm column (Phenomenex) in GF150 buffer (25 mM HEPES (pH 7.4), 150 mM KCl, and 1 mM MgCl$_2$) supplemented with 0.01% NP-40. Protein-containing fractions were pooled and supplemented with 0.1 mM ATP and 10% glycerol. For the KIF13B$^{CG}$ purification, cells were resuspended in 40 ml of PB buffer and were lysed by passing through an Emulsiflex C-3 (Avestin), clarified lysate was purified by Strep-Tactin XT resin, followed by anion exchange chromatography using a HiTrap Q column (Cytiva) equilibrated in HB buffer (35 mM PIPES−KOH pH 7.2, 1 mM MgSO$_4$, 0.2 mM EGTA, 0.1 mM EDTA, pH 7.4). Bound proteins were eluted with 45 mL of a linear gradient of 0–1 M NaCl in HB buffer. Eluted protein was further purified by superdex 200 columns (Cytiva)

in GF150 buffer. Protein-containing fractions were pooled and supplemented with 10% Glycerol. All Protein aliquots were subsequently flash frozen by liquid nitrogen and stored at −80 °C. Protein concentrations were measured by NanoDrop One (Thermo Fisher Scientific) based on the absorbance of the attached fluorophores.

Recombinant human tubulins (tubulin isotypes, α1A /βIII and α1A-Y /βIII in pFAST$^{TM}$-Dual vector) were expressed and purified using a modified protocol from previous studies[56,70]. Briefly, 1 ng of plasmid DNA was transformed into 50 µl of DH10Bac competent cells, positive colonies were verified by PCR and subcultured for bacmid DNA, and the recombinant tubulins were expressed in Sf9 cells using the same methods described for KIF13B proteins. To purify the recombinant tubulins, a pellet from 1 L Sf9 cells was thawed on ice and resuspended in 40 ml of lysis buffer (50 mM HEPES pH 7.2, 20 mM imidazole, 100 mM KCl, 1 mM MgCl$_2$) freshly supplemented with 0.5 mM 2-mercaptoethanol, 0.1 mM GTP, 1 mM phenylmethylsulfonyl fluoride (PMSF), and protease inhibitor mix (Promega). Resuspended cells were homogenized in a dounce homogenizer and lysed by the addition of 1% Triton X-100 and 5 µl of Benzonase Nuclease (Millipore) for 10 min on ice. To obtain soluble lysate, lysed cells were centrifuged at 15,000 × g for 20 min at 4 °C. The clarified lysate was filtered by a 0.22 mm filter (Millipore) and was loaded onto a 5 ml HisTrap HP columns (Cytiva), the protein was subsequently eluted by HisTrap Elution buffer (1X BRB80 plus 500 mM imidazole) freshly supplemented with 2 mM 2-mercaptoethanol, 0.2 mM GTP. Protein-containing fractions were pooled and diluted 3 times by 1 x PBS freshly supplemented with 0.2 mM GTP, then incubated with 2 ml of ANTI-FLAG® M2 affinity gel (Sigma-Aldrich) for 30 min at 4 °C and washed with 1×PBS. To elute the protein from the beads and remove the FLAG tag, the mixture was treated with PreScission protease ($V_{protein}$:$V_{protease}$ = 10:1) and incubated 2 h at 4 °C. Cleaved tubulin protein was subsequently eluted with 1×PBS freshly supplemented with 0.2 mM GTP. Protein was further purified by anion exchange chromatography using a HiTrap Q column (Cytiva) equilibrated in 1× BRB80 buffer freshly supplemented with 0.2 mM GTP. Bound proteins were eluted with a 25 ml of linear gradient of 0–1 M KCl in 1× BRB80 buffer freshly supplemented with 0.2 mM GTP. Eluted protein was diluted 10 times by 1× BRB80 buffer freshly supplemented with 0.2 mM GTP and then concentrated to -12.5 mg/ml using another 50 kDa MWCO centrifugal filter (Millipore). Recombinant tubulin concentrations were measured by absorption at 280 nm ($\epsilon_{280}$ = 115,000 M$^{-1}$ cm$^{-1}$ and MW = 100 kDa).

## Mass photometry

The mass photometry assays were carried out as previously described[50,60]. Briefly, to prepare chambers, microscope cover glass (#1.5 24 × 50 mm, Deckgläser) was cleaned by 1 h sonication in Milli-Q H$_2$O, followed by another hour sonication in isopropanol, cover glasses were then washed by Milli-Q H$_2$O and dried by filtered air. CultureWell silicone gaskets (Grace Bio-Labs) were cut and washed by Milli-Q H$_2$O, CultureWell silicone gaskets were then dried by filtered air and placed onto the freshly cleaned cover glasses providing four independent sample chambers. Samples and standard proteins were diluted in HP buffer (90 mM HEPES, 10 mM PIPES, 50 mM KCH$_3$COO, 2 mM Mg(CH$_3$COO)$_2$, 1 mM EGTA, 10% glycerol, pH = 7.0). The HP buffer was freshly filtered by a 0.22 µm filter before measurement. For calibration, standard proteins BSA (Sigma), Apoferritin (Sigma), and Thyroglobulin (Sigma) were diluted to 10–50 nM. The no-specific binding events of single molecules on cover glasses were scattered and the masses of these molecules were measured by the OneMP instrument (Refeyn) at room temperature. Data were collected at an acquisition rate of 1 kHz for 100 s by AcquireMP (Refeyn) and subsequently analyzed by DiscoverMP (Refeyn). For each concentration of recombinant proteins, the measurement was performed three times, and repeated with two different protein preparations. Results from one representative measurement are shown in the figure.

 

## Microtubule preparation

Porcine brain tubulin was isolated using the high-molarity PIPES procedure and then labeled with biotin NHS ester, Dylight-405 NHS ester, or Alexa647 NHS ester as described previously (https://mitchison.hms.harvard.edu/files/mitchisonlab/files/labeling_tubulin_and_quantifying_labeling_stoichiometry.pdf). Pig brains were obtained from a local abattoir and used within ~4 h. after death. To polymerize microtubules, 50 μM of unlabeled tubulin, 10 μM of biotin-labeled tubulin, and 3.5 μM of Dylight-405-labeled (or Alexa647-labeled) tubulin were incubated with 2 mM of GTP for 20 min at 37 °C. Polymerized microtubules were stabilized by the addition of 20 μM taxol and incubated additional 20 min. Microtubules were pelleted at 20,000×g by centrifugation over a 150 μl of 25% sucrose cushion and the pellet was resuspended in 50 μl BRB80 (80 mM Pipes (pH 6.8), 1 mM MgCl$_2$ and 1 mM EGTA) containing 10 μM taxol. The same method was used for microtubules polymerized from recombinant tubulin.

Carboxypeptidase A (CPA)-treated detyrosinated microtubules were prepared as per the previous description[29]. Briefly, 50 μM of unlabeled tubulin, 10 μM of biotin-labeled tubulin, and 3.5 μM of Alexa647-labeled microtubules tubulin were incubated with 2 mM of GTP and 12 mg/ml CPA (Sigma) for 20 min at 37 °C, followed by the addition of 20 μM taxol for an additional 20 min. The digestion was stopped by the addition of 10 mM DTT, and the CPA enzyme was removed by centrifugation of the microtubules over a 25% sucrose cushion and the pellet was resuspended as described above.

To prepare chimeric microtubules, stabilized tyrosinated and detyrosinated microtubules were equally mixed and incubated overnight at room temperature for end-to-end annealing. Annealed microtubules were verified by TIRF microscopy.

## Total internal reflection fluorescence (TIRF) assays

TIRF chambers were assembled from acid-washed glass coverslips as previously described (http://labs.bio.unc.edu/Salmon/protocolscoverslippreps.html), pre-cleaned slide, and double-sided sticky tape. Chambers were first incubated with 0.5 mg/ml PLL-PEG-biotin (Surface Solutions) for 5–10 min, followed by 0.5 mg/ml streptavidin for 5 min. Microtubules were diluted by BRB80 containing 10 μM taxol. Diluted microtubules were flowed into streptavidin-adsorbed flow chambers and incubated for 5–10 min at room temperature for adhesion. To remove unbound microtubules, chambers were subsequently washed twice with HP assay buffer (90 mM HEPES, 10 mM PIPES, 50 mM KCH$_3$COO, 2 mM Mg(CH$_3$COO)$_2$, 1 mM EGTA, 10% glycerol, bovine serum albumin (BSA) (1 mg/ml), biotin–BSA (0.05 mg/ml), K-casein (0.2 mg/ml), 0.5% Pluronic F-127 and 10 mM taxol, pH = 7.0). Purified motor protein was diluted to indicated concentrations in the assay buffer with 2 mM of corresponding nucleotides (ATP, ADP, or AMPPNP) and an oxygen scavenging system composed of PCA/PCD/Trolox. Then, the solution flowed into the glass chamber. Movies and images were acquired using a NIS-Elements software-controlled (AR, Version 5.20.2) Nikon Eclipse Ti2 Microscope (1.49 numerical aperture, ×100 objective) equipped with a TIRF illuminator and Andor iXon charge-coupled device electron multiplying camera. Data were analyzed manually using ImageJ (Fiji), and statistical tests were performed in GraphPad Prism 9.

## Reporting summary

Further information on research design is available in the Nature Portfolio Reporting Summary linked to this article.

## Data availability

All primary data used in this study are available within the figures and from the corresponding author upon request. Source data are provided with this paper.

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

## Acknowledgements
We thank all the members of the MOM lab for their continual input and feedback on this project. The authors thank Shinsuke Niwa (Tohoku University, Japan) for generously providing the purified VASH1–SVBP complex. The authors thank Antonina Roll-Mecak for generously providing the plasmids for recombinant human tubulin. The authors thank Kyoko Chiba for sharing the purified proteins KIF5B (1-912)-mScarlet-Strep, His-StrepII-sfGFP-HsKLC1, and HsKIF5B-PPS-mScarlet-StrepII. The authors thank Wenzhi Li for sharing the purified p150-N-term (1-530)-mScarlet-StrepII. The work was supported by a grant from NIGMS GM124889 (to R.J.M.).

## Author contributions
R.J.M. and X.F. designed the research. R.J.M. secured research funding. X.F. performed the research and analyzed the data. R.J.M. and X.F. wrote the paper. All authors edited the paper.

## Competing interests
The authors declare no competing interests.
