## [Peer Review File · Nature Communications]

REVIEWER COMMENTS

Reviewer #1 (Remarks to the Author):

The manuscript by Fan and McKenney investigates the role of the unique Cap-Gly domain of KIF13B and shows that it enhances both motor's landing rate and processive run lengths. Overall, the data in the manuscript is preliminary and fails to provide sufficient evidence to strongly make the case. Further, the manuscript does not provide physiological relevance to the Cap-Gly domain in KIF13B. The work is superficial and has insufficient experimental details to understand the role of the Cap-Gly domain and its importance in motor function. The model is vague and does not reflect the data. While I do not have a complete plan for proper characterization, here are some serious concerns:

- 1) Under recombinant KIF13B characterization, the authors claim that the oligomeric state of the full-length KIF13B motor is unclear. However, previous comprehensive work by Soppina et al. (2014) has elegantly demonstrated that, including KIF13B, members of kinesin-3 family motors exist in a monomeric state when not bound to cargo.
- 2) Although the mass photometry experiments were performed to characterize the oligomeric state of full-length and truncated versions of KIF13B motors, but failed to provide any new insights into the known oligomeric state of the motors.
- 3) The major claim of the paper is that KIF13B binds preferentially to tyrosinated microtubules. Although authors have performed in vitro motility assays to support the hypothesis, more rigorous characterization should be performed at the in vitro and cellular levels.
- 4) To truly establish the importance of the Cap-Gly domain as contributing factor to the preferential microtubule-binding and promote processive motility, a detailed mutational and biochemical analysis is required by generating motor chimeras.
- 5) Although KIF13B Δ P mutant has been shown to be a strong dimer in solution, authors used other mutants with weak dimerization potential for mass photometry and motility characterization.
- 6) As all the single-molecule motility analyses were performed at very low motor concentrations, using autoinhibited full-length and weak dimers to characterize the Cap-Gly domain can potentially cause problems while interpreting the findings.

7) The kymographs shown in Fig. 3A do not correlate with the author's explanation in the manuscript text. For example, authors claim that the KIF13B Δ tail had 2-3 fold short runs than full-length motors, but the kymographs shown in Fig. 3A clearly show longer runs for KIF13B Δ tail than the full-length motors.

8) In Fig. 4 and Fig. 5, the authors used full-length KIF13B motors and performed microtubule binding single molecule motility experiments. It is a well-known phenomenon that full-length wild-type motors exist in an autoinhibited state and do not interact with microtubules to prevent futile consumption of ATP. Nonetheless, the authors still used full-length motors to study the mechanism of motors binding to the microtubules in the presence of ADP and AMP-PNP, which is surprising and vague.

Together, the manuscript suffers from a number of serious flaws and the data does not support the authors conclusion and the significance of the Cap-Gly domain in KIF13B. The data is fluffy and has several loose ends. Therefore, I strongly believe that the manuscript does not meet the quality and standard of the journal and I do not recommend it for publication in Nature Communication

Reviewer #2 (Remarks to the Author):

Fan and McKenney studied how CAP-Gly domain that recognizes tyrosinated microtubules affect motility of kinesin KIF13B. The major finding of the paper is that it increases the run length of the motor from ~ 2 to ~ 6.4 microns by providing additional link between the motor and microtubules in the weakly bound ADP state of the motor. This is potentially an interesting paper with several interesting observations. I feel that understanding the oligomerization states of the motors and the relationship between their oligomeric states and processivity parameters (landing rate, run length and dwell time) are needed to fully reveal the molecular details of how Kif13B moves on microtubules.

Specific comments:

1. I do not quite understand the purpose of the Figure 2(B). It doesn't seem to have any extra information in addition to what is shown on Figure 3. SD analysis seems to make things more confusing. Generally, it may be useful for quantifying multiple quick on/off events, but under the conditions indicated on Fig 2 the motors are supposed to be moving processively as shown on Fig. 3. Processively moving motors will spend a lot of time on microtubules but would not show on SD map because they don't bind and unbind all the time. A better way to quantify single-molecule parameters seems to be what authors did do in Fig. 3 anyway. I don't think Fig. 2B adds much.

2. I would assume motors need to dimerize before they can move processively. Oligomeric state of the moving motors on Fig. 3A should be determined (e.g. by quantifying brightness, number of

fluorophores, etc.). Then histograms for the landing rate, lifetime and run length should be shown separately for monomers (is applicable), dimers, and higher oligomers (if any).

3. Figure 3B doesn't seem to be discussed in the text. What exactly does it show? Why does number of processively moving motors is always 1% for tyrosinated microtubules? Or is it somehow normalized? What do reduced numbers suggest? Are there less motors binding, or do some motors bind, but do not move?

4. What is the reason for using cumulative distributions on Fig. 3E and 5C? They seem a lot more counterintuitive and hinder analysis. Although mean is indicated as a number, eyeballing for example variance is difficult. I would recommend changing them to probability density (as done on Fig. 3D).

5. Related to the previous point. Run lengths on figure 3E are shown with four digits. It would be easier to see which digits are meaningful if it was a pdf, not cumulative distribution plot, so variance could be inferred. For example, once activated and moving, wt Kif13B dimer should have similar properties to V178Q and K414A (because it already has autoinhibition removed). Therefore, the run length of 6.3 for wild type is expected be no different from 5.7 for K414A. Is that true?

6. Figure 3G seems a bit trivial, doesn't it? Figure 3E shows that the motor can move processively on both tyrosinated and detyrosinated microtubules, albeit with different run length. Based on this, I would not expect motors to come off immediately as soon as they change from one lattice to another. So, figure 3G does make sense, but hardly makes any point.

7. Based on Fig. 4C Authors "hypothesize that avidity arising from the dimerization of the motor enhances the binding affinity of both the CAP-Gly and motor domains within Kif13B". This should be tested directly by imaging in single-molecule conditions similar to Fig.5 and counting how many fluorophores are there on microtubule bound motors. When ADP bound Kif13 interacts with microtubules is it a monomer or a dimer?

8. I agree that authors data on fig 4 indicate that "KIF13B's preference for tyrosinated microtubules is nucleotide dependent". However, that does not imply that the motor in different nucleotide states interacts differently with tyrosinated vs detyrosinated microtubules. This statement would assume that nucleotide state in some sort of way affects affinity of the CAP-Gly, which is difficult to image how that could be. More plausible explanation is that CAP-Gly adds to the affinity to all nucleotide states equally, but in ADP state it is more noticeable because the motor binding affinity is naturally weak.

9. The question from previous point could be addressed by imaging the behaviour of single CAP-Gly domains at single-molecule conditions (similar to Figure 5) to understand the landing rate, lifetime and diffusion coefficient. Knowing these parameters could be important to confirm authors conclusions about how this domain enhances affinity of the motor to the microtubule. Although authors indicate that their tail construct comes in different oligomeric states as suggested by mass photometry, direct imaging may help to understand the difference in behaviour between different oligomeric states.

10. Kymographs on Figure 5A show binding events with different brightness. Similar to the previous comment about fig 3, brightness should be quantified, so the parameters show below (landing rate and dwell time) could be attributed to monomers or dimers.

11. An informative way to reveal how oligomeric state of the motors affects its dynamics could be to make artificially stabilized dimers by introducing artificial dimerization domains (e.g. leucine zipper) and measuring its landing rates and processivity lengths.

Reviewer #3 (Remarks to the Author):

This manuscript investigates how the CAP-Gly domain of kinesin-3 KIF13B regulates its microtubule landing and subsequent motility on tyrosinated versus detyrosinated microtubules using in vitro single molecule motility assays. Several microtubule-associated proteins (such as the p150 subunit of dynein and CLIP170) contain a CAP-Gly domain that weakly interacts with the microtubules. Previous studies in the field showed that the CAP-Gly domain prefers binding to tyrosinated tail of alpha tubulin. This manuscript investigates the microtubule binding and motility of kinesin-3 KIF13B, the only kinesin motor with a C-terminal CAP-Gly domain at its tail region. The authors generated full-length and truncated KIF13B constructs and used carboxypeptidase A (or other relevant enzymes) to detyrosinate pig brain tubulin. In vitro single molecule motility assays revealed that KIF13B prefers to land on tyrosinated microtubules, and the construct that lacks the CAP-Gly domain (delta tail) has no preference to bind either microtubule, indicating that the CAP-Gly domain increases microtubule landing of the motor by interacting with the tyrosinated tail of alpha-tubulin. The authors have also performed measurements in the presence of 2 mM ADP and observed that KIF13B still prefers to bind to tyrosinated microtubules, indicating that when the motor domain is in a weakly-bound ADP state, microtubule binding is dominated by CAP-Gly binding to tyrosinated tubulin tail. The authors propose that KIF13B lands onto the microtubule through the interaction of its CAP-Gly domain with the tubulin tail, which facilitates the binding of the motor domain to the microtubule lattice, thereby playing a critical role in the initiation of transport.

Overall, the assays are well performed, and the results are broadly interesting to the researchers in the field. However, the work is performed using pig brain tubulin which is highly heterogeneous. There are now established methods to obtain homogenous tubulin in the field. In fact, the senior author of this manuscript has previously published a paper on dynein motility on tyrosinated versus detyrosinated microtubules polymerized from homogenous tubulin, and it is not clear to me why they decided to switch back to pig brain tubulin for this work. Collectively, I am enthusiastic about the publication of this work in Nature Communications, provided that the authors can repeat their measurements using recombinant tubulin.

Major Concerns

1. The experiments were performed using untreated versus enzymatically treated tubulin extracted from pig brains. The authors state that pig brain tubulin is 50% tyrosinated based on a previous publication. Detyrosinated microtubules are obtained by treating tubulin with carboxypeptidase A (CPA) or VASH1-SVBP. While they observe clear differences between treated versus untreated microtubules,

and they observe similar results from CPA- versus VASH1-SVBP- treated microtubules, these results need to be supported by mass spectrometry to the least.

2. Pig brain tubulin is highly heterogenous and is processed through a plethora of post-translational modifications. Recently, McKenney and other research groups have published papers for purifying recombinant tubulin and polymerizing homogenous microtubules with a well-defined tubulin post-translational modification. I strongly recommend the authors use recombinant tubulin to support their conclusions, as this would be the current state of the art in the field.

3. The authors compare the microtubule landing rate, velocity, run length, and residence times of a given motor construct on tyrosinated versus detyrosinated microtubules. While this is informative, they should also compare different motor constructs for the same preparation of microtubules. For example, in Figure 4B, the authors should titrate the motors and measure the apparent K_d for each condition and compare the K_d values of the constructs to each other on tyrosinated microtubules.

4. In their model, the authors wrote that "(1) In the ADP state, the CAP-Gly domain dominates the interaction between KIF13B and microtubules. Interaction between the CAP tyrosinated α -tubulin within microtubules facilitates the binding of the -Gly domain and the motor domain to the microtubule lattice." This is because ADP release is rate limited in physiological conditions and thereby kinesin motor domain mostly remains in the weakly-bound ADP state. While this is an attractive idea, it is important to know how strongly the kinesin motor domain versus the CAP-Gly domain interacts with the microtubule in 2 mM ADP. For that, the authors would need to perform K_d measurements in Figure 4B for the delta-motor construct and compare it to that of delta-tail. Based on the measurements performed at a single concentration in Figure 5B, the landing rate of delta-tail is two orders of magnitude higher than that of delta-motor. Therefore, it is possible that the motor domain more strongly interacts with microtubules than that of the tail even in the presence of ADP, therefore the CAP-Gly binding mode may not be that significant for initial landing onto the microtubule.

Minor Concerns

1. In Figure 1A, the results of the delta-motor construct are confusing. This is most likely due to aggregation or multimerization of this construct at higher concentrations, as stated by the authors. Have the authors considered truncating the tail further from its N-terminus? This may resolve the weird multimerization problem and enable them to study microtubule binding of the CAP-Gly domain in the absence of the motor domain (and part of the tail).

2. The SD analysis performed in Figure 2 is somewhat unusual, and I am not sure it adds more information than what we already learned from the analysis in Figure 3.

3. The authors wrote "KIF13B Δ tail had approximately two to three-fold shorter runs than full-length motors (Fig. 3E), which we hypothesize is possibly due to weaker dimerization of the motor at the low concentrations used for single molecule assays (Fig. 1C). However, we note that the run lengths for the full-length KIF13B are much longer than KIF13B Δ tail on tyrosinated microtubules (Fig. 3E), even though full length KIF13B is also predominantly monomeric at low concentrations (Fig. 1C). Therefore, it is possible that the full tail domain is required for stable dimeric assembly of KIF13B during processive motion, and more work, including structural analysis, is needed to fully address this question." Are they proposing that KIF13B may rapidly dimerize and monomerize on microtubules? While it is not

impossible, I found this to be unlikely. Later in the manuscript, the authors provide a more straightforward interpretation that the transient interactions between CAP-Gly and tubulin tail prevent full dissociation of the motor and enable reattachment of the motor domain to the microtubule. I would recommend deleting this section as it may be confusing to the reader.

4. The authors wrote, “we conclude that the KIF13B’s preference for tyrosinated microtubules is nucleotide-dependent.” This can be misunderstood as if the CAP-Gly interaction with the tubulin tail is nucleotide-dependent. I recommend a revision of this sentence.

5. The authors also wrote that CAP-Gly binding to the tubulin tail might stimulate dimerization and activation of processive motility. The authors do not present any evidence of dynamic dimerization and monomerization of KIF13B. Therefore, the sentence is too speculative as is and should be revised or removed from the manuscript.

6. While it is conceivable that p150 binding to EB1 at the plus end may localize the motor to the starting point of dynein-mediated transport, what could be the potential advantage of localizing a plus-end-directed KIF13B to the microtubule plus end through a similar mechanism?

We thank the reviewers for devoting their precious time towards reviewing our work and making helpful suggestions towards strengthening the manuscript in meaningful ways to improve its rigor and impact for the field. We have highlighted substantial text changes in our revision in blue colored font.

REVIEWER COMMENTS

We thank reviewer #1 for their time and consideration and hope that, although they were not enthusiastic about our results as first presented, they will consider our responses below within the full context of our work, its impact on the microtubule motor field, and more broadly, its impact on our understanding of kinesin motor regulation, an important topic for the cell biology community.

Reviewer #1 (Remarks to the Author):

The manuscript by Fan and McKenney investigates the role of the unique Cap-Gly domain of KIF13B and shows that it enhances both motor's landing rate and processive run lengths. Overall, the data in the manuscript is preliminary and fails to provide sufficient evidence to strongly make the case.

We respectfully disagree that our work is preliminary or lacks evidence to make our conclusions. We provide the first and only comprehensive biochemical and biophysical characterization of purified full-length KIF13B, reveal a previously unknown preference for tyrosinated microtubules, reveal the role of the CAP-Gly domain in this preference, and uncover an unexpected role of this interaction in KIF13B motility. All of our results provide a model for a novel role of the CAP-Gly domain, and the KIF13B tail domain more generally, in the regulation of motor activity, and unexpectedly, motility along microtubules. None of these results were previously known in the field, and we believe our experiments provide a comprehensive exploration of the molecular basis of these observed effects. We would ask the reviewer to enumerate which of our results they feel are preliminary so that we may more fully understand how to respond to this statement.

Further, the manuscript does not provide physiological relevance to the Cap-Gly domain in KIF13B. The work is superficial and has insufficient experimental details to understand the role of the Cap-Gly domain and its importance in motor function.

Evidence for a physiological role of the CAP-Gly domain in KIF13B function is provided in Mills et al. 2019 (PMCID: PMC6851494), who reported that deletion of the CAP-Gly domain of KIF13B results in subcellular mislocalization of the truncated motor, reduced LRP1-dependent endocytosis, and ultimately higher serum cholesterol levels due to defects in KIF13B trafficking. We have cited these results in our original discussion and now expanded on this point in our revised discussion. The goal of our work is to provide a biochemical and biophysical basis for why the CAP-Gly domain is physiologically important for KIF13B function. We have utilized our in vitro platform to interrogate the role of the CAP-Gly domain in motor movement with high detail, which we believe complements this existing data on the physiological importance of the CAP-Gly domain for KIF13B function in cells.

We respectfully disagree that our work is superficial or lacking in experimental details to understand the role of the CAP-Gly domain in motor function, as we have now described the first

and only known role for this domain in biasing the motor landing and processive movement along tyrosinated microtubules.

The model is vague and does not reflect the data. While I do not have a complete plan for proper characterization, here are some serious concerns:

1) Under recombinant KIF13B characterization, the authors claim that the oligomeric state of the full-length KIF13B motor is unclear. However, previous comprehensive work by Soppina et al. (2014) has elegantly demonstrated that, including KIF13B, members of kinesin-3 family motors exist in a monomeric state when not bound to cargo.

We agree that our results generally agree with the data from Soppina et al. with respect to the oligomeric state of the motor, as we cite in the results section. However, we disagree with the reviewer that the oligomeric state of the motor is settled in the field. We point out that our study is the first to utilize fully purified, full-length motors for this analysis, as Soppina et al. measured photobleaching steps from motors found in total cell lysates. We now mention this in the results section. Therefore, it is conceivable that other cellular factors influenced the oligomeric state of the motors in the experiments reported by Soppina et al. This is not the case with our fully purified motors, and our results report more accurately on the native state of the motor in the absence of any other cellular factors. Additionally, we extend the results reported by Soppina et al. by revealing that the oligomeric state of the motor is depends directly on the total motor concentration, as we observe increasing amounts of dimeric motors as we titrate the motor concentration in our mass photometry experiments (Fig. 1), even in the absence of cargo. Finally, as we mentioned in the results, other work showed presumably dimeric processive motion with tail truncated motors (Horiguchi et al, 2006, Duellberg et al, 2021), but further truncation of the tail domain results in predominantly monomeric motors (Ren et al. 2018). Thus, we believe the state of the native motor in cells is not settled, and our work provides new evidence on the behavior of the motor in isolation that will be valuable when considering models for how KIF13B behaves in a cellular context.

2) Although the mass photometry experiments were performed to characterize the oligomeric state of full-length and truncated versions of KIF13B motors, but failed to provide any new insights into the known oligomeric state of the motors.

We respectfully disagree. Our mass photometry results are the first and only measurements of the oligomeric state of purified full-length KIF13B motors. More importantly, they clearly reveal that the oligomeric state of wild-type full-length or truncated motors is concentration-dependent. We note this effect is observed in the absence of any cargo, in contrast to current models in the field that the motor must bind to cargo in order to dimerize.

3) The major claim of the paper is that KIF13B binds preferentially to tyrosinated microtubules. Although authors have performed in vitro motility assays to support the hypothesis, more rigorous characterization should be performed at the in vitro and cellular levels.

We believe our in vitro analysis is rigorous and are unclear why the reviewer thinks not. We fully agree that analysis of KIF13B's preference for tyrosinated microtubules in vivo is a clear and logical next step. We are attempting experiments in vivo in the lab, but believe the work involved to deconvolute the dynamic cellular localization of KIF13B and dynamically modified microtubules warrants its own independent investigation. Our current work focuses on the discovery and

subsequent characterization of the molecular and biophysical basis for this effect using well-defined *in vitro* reconstitutions. We believe these reconstitutions provide the most rigorous experimental platform with which to investigate these questions, as they are defined systems, free from the unknown variables of the intracellular milieu. Our *in vitro* experiments make defined predictions that warrant follow-up investigations *in vivo*, but we believe these investigations go well-beyond the scope of a single paper and are better suited for their own full study.

4) To truly establish the importance of the Cap-Gly domain as contributing factor to the preferential microtubule-binding and promote processive motility, a detailed mutational and biochemical analysis is required by generating motor chimeras.

We have performed a detailed mutagenesis analysis by creating multiple construct that contain or lack the CAP-Gly domain. In response to this suggestion, we now include a new chimeric motor that consists of the orthogonal KIF5B motor fused to the KIF13B CAP-Gly domain. We made this chimeric motor to examine if the insertion of an exogenous CAP-Gly domain to the C-terminal tail domain of an orthogonal kinesin was effective at biasing that motor towards tyrosinated microtubules, as we observe for KIF13B. Indeed, we see robust targeting of our chimeric motor to tyrosinated microtubules, further supporting our data that the C-terminal domain of a kinesin can strongly bias the landing and motile prosperities of the motor towards specific subsets of microtubules. This new data is now included in new Fig. S5, and we believe it strongly supports our model whereby specialized kinesin tail domains dictate the motile parameters of the motor. We thank the reviewer for this excellent suggestion that strengthens our work and conclusions.

5) Although KIF13B Δ P mutant has been shown to be a strong dimer in solution, authors used other mutants with weak dimerization potential for mass photometry and motility characterization.

During the initial phases of this project, we did produce the KIF13B Δ P motor for analysis. However, we found that this mutant was prone to degradation and aggregation during purification and in our motility assays, and therefore unsuitable for further detailed analysis. We note that the aggregates of this mutant were often motile, consistent with high activity in cells reported in Soppina et al. 2014. We included example data of a KIF13B Δ P purification and single molecule assay here. The more prominent degradation is apparent in the gel filtration fractions, along with the shift of the majority of the protein into the void volume of the column, as compared to our KIF13B^{K414A} motor (Reviewer Fig. 1). The example image from our TIRF assay clearly shows the prominent aggregation and motility of aggregated motors. Further, the mutants that we did choose to study (K414A and V178Q) were structurally characterized at atomic resolution by Ren et al. 2018, lending further confidence in using these mutants in our study.

Reviewer Figure. 1: Left: SDS-PAGE gel of gel filtration elution fractions showing degradation of the KIF13B Δ P protein (red arrow denotes full-length motor) along with shift of the motor into the void volume of the column (fraction 8) as compared to the KIF13B^{K414A} motor shown below. Right: TIRF image showing KIF13B Δ P particles bound to and moving along a microtubule. Note the prominently brighter aggregates which both bind statically and move processively along the microtubule.

Consistent with that study, we show here that these mutations do not majorly perturb the structural organization of the motor leading to aggregation like we observed with the KIF13B^{ΔP} motor variant.

6) As all the single-molecule motility analyses were performed at very low motor concentrations, using autoinhibited full-length and weak dimers to characterize the Cap-Gly domain can potentially cause problems while interpreting the findings.

It is unclear to us what specific problems the reviewer is concerned about. We set out to characterize minimally perturbed, native proteins in the effort to understand biological function. If the reviewer prefers to analyze artificially dimerized or artificially truncated motors (as is common in the field), we would argue that the introduction of non-native, forced oligomeric states is more problematic for understanding the native behavior of the motor than is using a native motor with no artificial engineering. In this context, we note that we do include direct comparisons of the native, unperturbed motor, with artificially activated (and as minimally perturbed as possible with single point mutants) motors in the same assays. Further, using wild-type motors allowed us to observe the behavior of the CAP-Gly domain in the most native state possible in vitro. Rather than problematic, we view our efforts to keep the motor in the most native state possible as critically important for our conclusions about the role of the CAP-Gly domain in KIF13B motor activity.

7) The kymographs shown in Fig. 3A do not correlate with the author's explanation in the manuscript text. For example, authors claim that the KIF13B^{Δtail} had 2-3 fold short runs than full-length motors, but the kymographs shown in Fig. 3A clearly show longer runs for KIF13B^{Δtail} than the full-length motors.

While we can observe longer runs for the KIF13B^{Δtail} construct as mentioned by the reviewer, the quantification of the motor's behavior clearly revealed that the average run length was much shorter than for full-length motors. We have replaced the examples kymographs for this construct with new ones that show more clearly the individual runs of KIF13B^{Δtail} to make this point clearer for the reader. The original kymographs were very dense, possibly obscuring the shorter runs.

8) In Fig. 4 and Fig. 5, the authors used full-length KIF13B motors and performed microtubule binding single molecule motility experiments. It is a well-known phenomenon that full-length wild-type motors exist in an autoinhibited state and do not interact with microtubules to prevent futile consumption of ATP. Nonetheless, the authors still used full-length motors to study the mechanism of motors binding to the microtubules in the presence of ADP and AMP-PNP, which is surprising and vague.

Please see our comments above about using native motor constructs whenever possible to make the most accurate conclusions about native motor behavior. Obviously, full-length native motors must and do interact with microtubules within cells, and we view our experiments with the full-length motors in these experiments as attempts to analyze such interactions in detail. We agree that the motors exist in an autoinhibited state, but clearly that conformation can and must be relieved in order for motor movement along microtubules, and we suggest that the binding to microtubules may play a critical role in this process. Indeed, in the case of KIF13B and our new KIF5B^{CG} chimeric motor, our data clearly reveal that the CAP-Gly domain plays a prominent role in the motor-microtubule interaction when the motor domain is bound to ADP, and thus locked in

a weak (though non-zero) affinity state (Fig. 4B, S5). Our data also clearly show that when the motor domain is bound to ATP (AMP-PNP), that the motor domain is capable of strongly interacting with the microtubule and overpowering the tyrosination preference of the CAP-Gly domain (Fig. 4B). Thus, our data reveal that the autoinhibition state of the motor is NOT sufficient to prevent the motor domain-microtubule interaction in this specific nucleotide state. Rather than vague, we believe these results provide much needed clarity to the field about the role of each microtubule-binding domain in the KIF13B-microtubule interaction at different critical states of the nucleotide hydrolysis cycle. These insights are even more important as they were made with the physiologically relevant full-length constructs that allowed for a direct comparison of the motor domain vs. CAP-Gly domain within the same polypeptide, as is relevant in vivo.

Together, the manuscript suffers from a number of serious flaws and the data does not support the authors conclusion and the significance of the Cap-Gly domain in KIF13B. The data is fluffy and has several loose ends. Therefore, I strongly believe that the manuscript does not meet the quality and standard of the journal and I do not recommend it for publication in Nature Communication

We disagree with the conclusion of the reviewer that our data does not support our conclusions. However, we are unable to respond fully to non-specific comments about data interpretation. Further, we feel our data provides a comprehensive analysis of the two major microtubule-binding domains within KIF13B and provides molecular insight into the coordination and outcomes of these domains as they are relevant for KIF13B motility along microtubules. If the reviewer has more specific comments about which data contain “Fluff” or “loose ends”, we would be happy to respond.

Reviewer #2 (Remarks to the Author):

Fan and McKenney studied how CAP-Gly domain that recognizes tyrosinated microtubules affect motility of kinesin KIF13B. The major finding of the paper is that it increases the run length of the motor from ~2 to ~6.4 microns by providing additional link between the motor and microtubules in the weakly bound ADP state of the motor. This is potentially an interesting paper with several interesting observations. I feel that understanding the oligomerization states of the motors and the relationship between their oligomeric states and processivity parameters (landing rate, run length and dwell time) are needed to fully reveal the molecular details of how Kif13B moves on microtubules.

Specific comments:

1. I do not quite understand the purpose of the Figure 2(B). It doesn't seem to have any extra information in addition to what is shown on Figure 3. SD analysis seems to make things more confusing. Generally, it may be useful for quantifying multiple quick on/off events, but under the conditions indicated on Fig 2 the motors are supposed to be moving processively as shown on Fig. 3. Processively moving motors will spend a lot of time on microtubules but would not show on SD map because they don't bind and unbind all the time. A better way to quantify single-molecule parameters seems to be what authors did do in Fig. 3 anyway. I don't think Fig. 2B adds much.

Thank you for your helpful review. The SD analysis performed here is an easy and convenient way to determine preferential interactions with microtubules, and has been used in past studies of microtubule motor movement both in vivo and in vitro (Cai et al. Biophys J. 2007, Cai et al.

PLOS Biology. 2009, and McKenney et al. EMBO J. 2016), which we have cited in this section. While the reviewer is correct that processive motors spend more time on microtubules, over the course of the entire move, the fluorescence intensity still fluctuates much more than background as shown in our data and data from the citations above. In addition, the analysis is useful to quantify non-motile interactions with microtubules, as with KIF13B^{CG} shown in Figure 2B. We believe Figure 2 provides an important introductory observation that, in aggregate, the KIF13B motors strongly prefer to interact with tyrosinated microtubules in some manner, although all those interactions may not necessarily be processive movements. This observation sets the stage for the in-depth analysis we subsequently perform in the following figures. The data in Fig. 2 do not delineate between rapid binding kinetics or longer processive interactions, which requires more in-depth analysis shown in Fig. 3 to determine which kinetic parameters are affected by tyrosination. This is why we end that section of the results with *“Because the KIF13B motor domain is not affected by the tyrosination state of the lattice (Fig. 2B), these results reveal that the CAP-Gly domain enhances the initial microtubule encounter rate, interaction time of the motor on tyrosinated microtubules, or both.”* We have also used this assay to determine the effects of the tyrosination state of the lattice using recombinant human tubulin in our revised manuscript (New Fig. 2C). Because recombinant tubulin is very difficult to prepare, this assay is ideal to utilize when limiting amounts of reagents are available. Therefore, revised Fig. 2 contains more novel information than in our initial submission. We believe that Fig. 2 is an important introductory observation for our story, and would prefer to leave it in the manuscript. However, if the reviewer feels very strongly that this figure detracts from the readability of our paper, we can move it to the supplement.

2. I would assume motors need to dimerize before they can move processively. Oligomeric state of the moving motors on Fig. 3A should be determined (e.g. by quantifying brightness, number of fluorophores, etc.). Then histograms for the landing rate, lifetime and run length should be shown separately for monomers (is applicable), dimers, and higher oligomers (if any).

We also believe the motors must be dimers in order to move processively, and we now include new analysis in our revised manuscript to quantify the oligomerization state of the motors in our assays. In new Fig. S2, we provide two pieces of data to confirm the oligomerization state of the motors when moving processively in the presence of ATP or when interacting transiently with microtubules in the ADP state. First, we co-expressed orthogonally tagged motors and examined their behavior in our single molecule assay. Our new results definitively show that an expected fraction of motors is dual-labeled with both fluorescent tags when moving processively or when binding to microtubules in ADP. These results directly demonstrate that the motors oligomerize in order to move processively (as expected), and reveal oligomerization of the motors interacting transiently with microtubules in ADP.

To build on these observations, we also measured the GFP intensity of motor constructs in the presence of ADP on microtubules and compared them to the well-characterized dimeric KIF5B motor as an intensity standard (Fig. S2). This analysis also revealed that the intensity of processive motors is very similar to the intensity of the dimeric KIF5B motor, and further reveals that the truncated CAP-Gly domain appears to dimerize at low concentration when bound to microtubules. The latter result is surprising and may reveal a previously unknown regulatory role for the CAP-Gly and unstructured tail domain in dimerization of the motor, but further research is necessary to confirm this idea. We thank the reviewer for their input and valuable suggestion that makes our results more impactful!

3. Figure 3B doesn't seem to be discussed in the text. What exactly does it show? Why does number of processively moving motors is always 1% for tyrosinated microtubules? Or is it somehow normalized? What do reduced numbers suggest? Are there less motors binding, or do some motors bind, but do not move?

We apologize this wasn't clear. In the text, we state *"The landing rate of KIF13B was more than 50-fold less than that of truncated or mutant motors (Fig. 3B), reflecting the strong autoinhibition of the wild-type motor⁵², and its predominantly monomeric state (Fig. 1C)."* whereby we describe the effects of tyrosination/detyrosination on the landing rate of the motors onto microtubules. The graphs in Fig. 3C are normalized to the landing rate of tyrosinated microtubules within the same chambers as the detyrosinated microtubules and we have updated the graph axis label to *"No. of Processively Moving (Fraction of Tyr MTs)"* to more clearly state this. We apologize for the confusion, but the 1.0 on these graphs represents 100% and not 1%.

4. What is the reason for using cumulative distributions on Fig. 3E and 5C? They seem a lot more counterintuitive and hinder analysis. Although mean is indicated as a number, eyeballing for example variance is difficult. I would recommend changing them to probability density (as done on Fig. 3D).

Cumulative frequency fitting is commonly used in the field to avoid artificially binning the data and introducing bias into the mathematical fits to the run-length measurements as described originally (we believe) by Thorn and Vale, JCB 2000. This procedure fits all the data directly (as opposed to binning), is widely used in the field, and we prefer to use it as we have done in our previous studies.

5. Related to the previous point. Run lengths on figure 3E are shown with four digits. It would be easier to see which digits are meaningful if it was a pdf, not cumulative distribution plot, so variance could be inferred. For example, once activated and moving, wt Kif13B dimer should have similar properties to V178Q and K414A (because it already has autoinhibition removed). Therefore, the run length of 6.3 for wild type is expected be no different from 5.7 for K414A. Is that true?

Thank you for pointing this out. We have rounded these numbers to the nearest tenth for clarity. The reviewer is correct that the numbers for WT vs the activated mutant motors are very similar and we do claim a statistical difference. We do not believe that small variations in run-lengths are physiologically relevant, and likely attributable to experimental variation as the reviewer suggests. These numbers are useful for broad comparison between the tyrosinated and detyrosinated microtubules within the same experiment, i.e. the WT motor moves approximately 3-fold longer on tyrosinated vs. detyrosinated microtubules within the same experiment.

6. Figure 3G seems a bit trivial, doesn't it? Figure 3E shows that the motor can move processively on both tyrosinated and detyrosinated microtubules, albeit with different run length. Based on this, I would not expect motors to come off immediately as soon as they change from one lattice to another. So, figure 3G does make sense, but hardly makes any point.

We believe Fig. 3E provides an important demonstration that the CAP-Gly domain does not need to be continuously engaged with the microtubule lattice during processive motility. The reviewer is correct that once motility commences on detyrosinated microtubules, it appears to proceed relatively unimpeded, but we wanted to confirm this point in an alternative manner using the

experiment shown in revised Fig. 3F-G. The data lead us to suggest that the CAP-Gly domain may prevent detachment of the motor during motility through rapid re-binding kinetics upon motor domain detachment, as we suggest in the following sentences.

7. Based on Fig. 4C Authors “hypothesize that avidity arising from the dimerization of the motor enhances the binding affinity of both the CAP-Gly and motor domains within Kif13B”. This should be tested directly by imaging in single-molecule conditions similar to Fig.5 and counting how many fluorophores are there on microtubule bound motors. When ADP bound Kif13 interacts with microtubules is it a monomer or a dimer?

Please see our response to point #2 above and our new data in Fig. S2 revealing the full-length motor and the isolated CAP-Gly domains appear to dimerize when bound to microtubules in the presence of ADP.

8. I agree that authors data on fig 4 indicate that “KIF13B’s preference for tyrosinated microtubules is nucleotide dependent”. However, that does not imply that the motor in different nucleotide states interacts differently with tyrosinated vs detyrosinated microtubules. This statement would assume that nucleotide state in some sort of way affects affinity of the CAP-Gly, which is difficult to image how that could be. More plausible explanation is that CAP-Gly adds to the affinity to all nucleotide states equally, but in ADP state it is more noticeable because the motor binding affinity is naturally weak.

We agree that our wording may lead to some confusion and have re-worded this conclusion. We thank the reviewer for their help with this point.

9. The question from previous point could be addressed by imaging the behaviour of single CAP-Gly domains at single-molecule conditions (similar to Figure 5) to understand the landing rate, lifetime and diffusion coefficient. Knowing these parameters could be important to confirm authors conclusions about how this domain enhances affinity of the motor to the microtubule. Although authors indicate that their tail construct comes in different oligomeric states as suggested by mass photometry, direct imaging may help to understand the difference in behaviour between different oligomeric states.

Thank you for this suggestion. We have now replaced our previous tail domain construct with a more truncated CAP-Gly domain construct, KIF13B^{CG}. We fully expected this truncation to be monomeric, but to our surprise, we could see concentration dependent dimerization in our mass photometry assay (Fig. 1) and single molecule brightness analysis, as requested by the reviewer, confirmed that the construct appears to dimerize when bound to microtubules (Fig. S2). We believe these new results may hint at further regulation of the KIF13B oligomeric state, which is ripe for future studies. We analyzed the landing rates and dwell time distribution for KIF13B^{CG} (which does not bind to detyrosinated microtubules) and observe very consistent results as the reviewer predicted, that the CAP-Gly domain in isolation adds an extra contact point of affinity for the overall KIF13B motor on tyrosinated microtubules. For instance, the relative dwell times of full-length KIF13B is approximated by the sum of the individual dwell times of KIF13B^{CG} and KIF13B^{Δtail} (Fig. 5). We have incorporated the reviewer’s suggestion for clarity on this topic in our results section discussing these results.

10. Kymographs on Figure 5A show binding events with different brightness. Similar to the previous comment about fig 3, brightness should be quantified, so the parameters shown below (landing rate and dwell time) could be attributed to monomers or dimers.

We have provided brightness distribution analysis for our activated motor construct and our new CAP-Gly domain construct in new Fig. S2. The analysis reveals that the majority of activated full-length motors are dimers in ADP. We replaced our previous delta-motor construct, which formed spurious oligomers in solution, with a new more minimal CAP-Gly domain construct (New Fig. 1). Surprisingly, our mass photometry and brightness analysis revealed that our new CAP-Gly domain construct is predominantly monomeric in solution, but dimeric when interacting with microtubules. This is an interesting new point that deserves further investigation, as we now point out in the manuscript. Finally, we performed a two-color experiment whereby co-expression of differentially labeled motors was used to assess the oligomeric state of the motors in our assays. The results in new Fig. S2 show a substantial fraction of two-color motors in both ATP and ADP conditions, suggesting that the majority of activated motors are dimers in these conditions, which is also supported by the new intensity analysis.

11. An informative way to reveal how oligomeric state of the motors affects its dynamics could be to make artificially stabilized dimers by introducing artificial dimerization domains (e.g. leucine zipper) and measuring its landing rates and processivity lengths.

While we agree about the usefulness of artificially dimerized constructs, we note that most publications in the field have done precisely this, and made conclusions about native motor behavior based on these artificial constructs. One of our goals was to analyze native motor constructs to provide an accurate report on the biophysical behaviors of near native motors.

Reviewer #3 (Remarks to the Author):

This manuscript investigates how the CAP-Gly domain of kinesin-3 KIF13B regulates its microtubule landing and subsequent motility on tyrosinated versus detyrosinated microtubules using in vitro single molecule motility assays. Several microtubule-associated proteins (such as the p150 subunit of dynactin and CLIP170) contain a CAP-Gly domain that weakly interacts with the microtubules. Previous studies in the field showed that the CAP-Gly domain prefers binding to tyrosinated tail of alpha tubulin. This manuscript investigates the microtubule binding and motility of kinesin-3 KIF13B, the only kinesin motor with a C-terminal CAP-Gly domain at its tail region. The authors generated full-length and truncated KIF13B constructs and used carboxypeptidase A (or other relevant enzymes) to detyrosinate pig brain tubulin. In vitro single molecule motility assays revealed that KIF13B prefers to land on tyrosinated microtubules, and the construct that lacks the CAP-Gly domain (delta tail) has no preference to bind either microtubule, indicating that the CAP-Gly domain increases microtubule landing of the motor by interacting with the tyrosinated tail of alpha-tubulin. The authors have also performed measurements in the presence of 2 mM ADP and observed that KIF13B still prefers to bind to tyrosinated microtubules, indicating that when the motor domain is in a weakly-bound ADP state, microtubule binding is dominated by CAP-Gly binding to tyrosinated tubulin tail. The authors propose that KIF13B lands onto the microtubule through the interaction of its CAP-Gly domain with the tubulin tail, which facilitates the binding of the motor domain to the microtubule lattice, thereby playing a critical role in the initiation of transport.

Overall, the assays are well performed, and the results are broadly interesting to the researchers in the field. However, the work is performed using pig brain tubulin which is highly heterogenous. There are now established methods to obtain homogenous tubulin in the field. In fact, the senior author of this manuscript has previously published a paper on dynein motility on tyrosinated versus detyrosinated microtubules polymerized from homogenous tubulin, and it is not clear to me why they decided to switch back to pig brain tubulin for this work. Collectively, I am enthusiastic about the publication of this work in Nature Communications, provided that the authors can repeat their measurements using recombinant tubulin.

We thank the reviewer for their careful reading of our paper and for recognizing the impact of our work on the field. While several labs have published methods for producing recombinant tubulin using baculovirus expression, this methodology is difficult, expensive, and does not yield large amounts of material. In McKenney et al. 2016, we utilized a yeast expression system that also did not yield large amounts of material, making the experiments more difficult and time consuming. This system was utilized in the Vale laboratory, which has substantially more resources available than we currently have in the McKenney lab. We also note that in that study, detyrosination of brain tubulin using CPA was utilized in parallel and showed similar results as recombinant detyrosinated microtubules.

Detyrosination of brain tubulin by CPA, and more recently by the tubulin detyrosinating enzyme VASH, is a method that has been used in dozens of prior studies, and thus has a great track record of use and reproducibility in the field. We show here that utilizing CPA or VASH on brain tubulin gives similar results with respect to KIF13B microtubule preference (Supp. Fig. 3,4). Thus, we believe that our utilization of CPA to generate detyrosinated microtubules is broadly supported by prior studies in the field and represents a highly validated experimental approach to study the effects of tyrosination on microtubule-associated proteins. Nonetheless, it is true that brain tubulin contains a mixture of tubulin isoforms and other PTMs, which may be a concern in our study. We have spent a substantial amount of effort to generate recombinant human tubulin in sufficient yields to respond to this request. We encountered many hurdles in the production of substantial amounts of active, recombinant tubulin that reflects the difficulty of working with this reagent. In line with this, we note that only a handful of very well-funded labs in the field are currently routinely publishing studies using recombinant tubulin (Roll-Mecak, Surrey, Kapoor labs). Our lab has only a fraction of the resources of these highly funded, well-established labs, making the production of large amounts of recombinant tubulin prohibitive for us. We were able to produce minimal amounts of active recombinant tubulin and we tested it in our differential binding assay with both our control CAP-Gly domain protein, along with our panel of KIF13B constructs (New Fig. 2C). Comfortingly, we observe the same strong differential binding of KIF13B to tyrosinated versus detyrosinated microtubules made from recombinant human tubulin, lending even stronger support to our prior observations on brain microtubules detyrosinated with either CPA or VASH-SVP. Thus, we have validated the differential binding of KIF13B on detyrosinated microtubules made by three independent methods (CPA, VASH-SVP, recombinant tubulin) and found consistent results, which we believe goes well above and beyond the standard of most publications in the field. We hope these results and our efforts now convince the reviewer that the tyrosination state of the microtubule, regardless of tubulin isotype or orthogonol PTMs, is the primary driver for differential KIF13B binding.

Major Concerns

1. The experiments were performed using untreated versus enzymatically treated tubulin extracted from pig brains. The authors state that pig brain tubulin is 50% tyrosinated based on a previous publication. Detyrosinated microtubules are obtained by treating tubulin with carboxypeptidase A (CPA) or VASH1-SVBP. While they observe clear differences between treated versus untreated microtubules, and they observe similar results from CPA- versus VASH1-SVBP- treated microtubules, these results need to be supported by mass spectrometry to the least.

Please see our comments above. Utilizing CPA to generate detyrosinated microtubules has been a standard method since the 1970's (Raybin and Flavin, JCB 1978), and subsequently used in dozens of studies. Thus, it is firmly established that CPA digestion removes the terminal tyrosine from α -tubulin. In addition, we show data that generation of detyrosinated microtubules with the more recently identified bona-fide detyrosinating enzyme VASH/SVP (Aillaud et al. and Nieuwenhuis et al. 2017) gives highly similar results, further bolstering support for the specific effects of CPA on microtubules. We do not believe our study should be held to a higher arbitrary standard than the rest of the field for the last \sim 50 years and feel that our data with detyrosinated brain tubulin present a compelling case for the effects of detyrosination on KIF13B activity. However, we have produced and tested recombinant human tubulin in response to this request and have shown that KIF13B similarly discriminates between tyrosinated and detyrosinated tubulin as we reported for both CPA and VASH treated brain tubulin (new Fig. 2C). We believe these new experiments further strengthen our conclusion that the tyrosination state of the lattice alone strongly dictates the interaction between KIF13B and microtubules.

2. Pig brain tubulin is highly heterogenous and is processed through a plethora of post-translational modifications. Recently, McKenney and other research groups have published papers for purifying recombinant tubulin and polymerizing homogenous microtubules with a well-defined tubulin post-translational modification. I strongly recommend the authors use recombinant tubulin to support their conclusions, as this would be the current state of the art in the field.

Please see our response to this request above and our new data with recombinant human tubulin in new Fig. 2C.

3. The authors compare the microtubule landing rate, velocity, run length, and residence times of a given motor construct on tyrosinated versus detyrosinated microtubules. While this is informative, they should also compare different motor constructs for the same preparation of microtubules. For example, in Figure 4B, the authors should titrate the motors and measure the apparent K_d for each condition and compare the K_d values of the constructs to each other on tyrosinated microtubules.

We thank the reviewer for this suggestion and agree that measured K_D values would be a valuable insight. In fact, we initially did this experiment prior to submission but found that the affinities of the WT motor are so low for detyrosinated microtubules in ADP, that we are not confident in the fit of the data and resulting K_D values. We provide the reviewer the data in the graphs below (Reviewer Fig. 2), which shows the very poor binding of the WT motor to detyrosinated MTs. For this reason, we chose to only show quantification of the amount of binding in the highest concentration we tested in the main figure. In the case of the activated KIF13B^{K414A} motor, the binding affinity is improved due to release from autoinhibition and enhanced dimerization. In this case, we measure an approximately 3-fold increase in affinity for tyrosinated versus detyrosinated microtubules. We now include the data for KIF13B^{K414A} in the text of our results section.

Reviewer Figure. 2: Fluorescence intensity data and fits to the Michaelis-Menten equation to derive the K_D of the indicated constructs for microtubules. Left: The WT KIF13B motor does not bind with high enough affinity to detyrosinated microtubules to derive a reliable K_D measurement. Right: The activated KIF13B^{K414A} mutant shows higher affinity for both tyrosinated and detyrosinated microtubules, as expected.

4. In their model, the authors wrote that “(1) In the ADP state, the CAP-Gly domain dominates the interaction between KIF13B and microtubules. Interaction between the CAP tyrosinated α -tubulin within microtubules facilitates the binding of the CAP-Gly domain and the motor domain to the microtubule lattice.” This is because ADP release is rate limited in physiological conditions and thereby kinesin motor domain mostly remains in the weakly-bound ADP state. While this is an attractive idea, it is important to know how strongly the kinesin motor domain versus the CAP-Gly domain interacts with the microtubule in 2 mM ADP. For that, the authors would need to perform K_D measurements in Figure 4B for the delta-motor construct and compare it to that of delta-tail. Based on the measurements performed at a single concentration in Figure 5B, the landing rate of delta-tail is two orders of magnitude higher than that of delta-motor. Therefore, it is possible that the motor domain more strongly interacts with microtubules than that of the tail even in the presence of ADP, therefore the CAP-Gly binding mode may not be that significant for initial landing onto the microtubule.

Thank you for this informative thought. First, we believe the CAP-Gly domain interaction with microtubules is very significant for the initial landing onto the microtubule, as we observe large differential interactions between tyrosinated versus detyrosinated microtubules with full length motors, but not with tail truncated motors (Figs. 2-3). We don't see another way to interpret that data other than in the context of the full-length motor, the CAP-Gly domain dictates the landing

onto tyrosinated microtubules. The key point may be in the context of the full-length motor, in which the motor domains may be in an autoinhibited state that reduces their ability to interact with the microtubule (as is observed in the structures of tail truncated KIF13B and the recent structure of full-length KLP-6). We have measured the single molecule landing rates and dwell times for our new KIF13B^{CG} construct (a dimer), along with the KIF13B^{Δtail} construct in ADP, which indeed reveals a much higher landing rate for KIF13B^{Δtail}. In addition, the KIF13B^{Δtail} construct shows longer dwell times in ADP than KIF13B^{CG}, consistent with the reviewer's assertion that the ADP-motor domains have a higher affinity for microtubules than the CAP-Gly domain. However, the caveat to these experiments is in using truncated constructs that do not recapitulate the endogenous folded structure of the full-length KIF13B molecule. Indeed, the full-length motor shows landing rates of the same order of magnitude as the isolated CAP-Gly domain, and an order of magnitude less than the KIF13B^{Δtail} construct, suggesting that the intramolecular folding/autoinhibition may strongly attenuate the interaction between the motor domains and microtubule. Supporting this idea, the activated KIF13B^{K414A} motor shows landing rates on the same order of magnitude as the truncated KIF13B^{Δtail}. However, the dwell times of the full-length KIF13B are longer than KIF13B^{CG}, suggesting to us that once the motor engages on the microtubule, the ADP-motor domains may contribute to the interaction, possibly through microtubule-stimulated exchange of the bound ADP. Finally, we note that the dwell times of KIF13B and KIF13B^{K414A} are similar on the same types of microtubules, further indicating similar oligomeric states and contributions of both the CAP-Gly and motor domains.

We have updated our results discussion to address this point and thank the reviewer for their insight.

Minor Concerns

1. In Figure 1A, the results of the delta-motor construct are confusing. This is most likely due to aggregation or multimerization of this construct at higher concentrations, as stated by the authors. Have the authors considered truncating the tail further from its N-terminus? This may resolve the weird multimerization problem and enable them to study microtubule binding of the CAP-Gly domain in the absence of the motor domain (and part of the tail).

Thank you for this suggestion. We have replaced the previous delta-motor construct with a much shorter region that encompasses just the CAP-Gly domain and the remaining unstructured C-terminus (new Fig. S1). Surprisingly, we find even this small region is capable of concentration-dependent dimerization in solution (new Fig. 1). and appears to dimerize when bound to microtubules (new Fig. S2). We believe this construct is a better tool to study the isolated behavior of the CAP-Gly domain, which is dimeric in the motor's activated state in cells (Soppina et al. 2014, and our data here). We have performed all new measurements using this construct in our revised manuscript.

2. The SD analysis performed in Figure 2 is somewhat unusual, and I am not sure it adds more information than what we already learned from the analysis in Figure 3.

Thank you for this comment. Please see our response to Reviewer #2 above about this figure. This figure now contains our new data with recombinant human tubulin, as requested by this reviewer, providing more novel information in the figure than the initial submission.

3. The authors wrote “KIF13B Δ tail had approximately two to three-fold shorter runs than full-length motors (Fig. 3E), which we hypothesize is possibly due to weaker dimerization of the motor at the low concentrations used for single molecule assays (Fig. 1C). However, we note that the run lengths for the full-length KIF13B are much longer than KIF13B Δ tail on tyrosinated microtubules (Fig. 3E), even though full length KIF13B is also predominantly monomeric at low concentrations (Fig. 1C). Therefore, it is possible that the full tail domain is required for stable dimeric assembly of KIF13B during processive motion, and more work, including structural analysis, is needed to fully address this question.” Are they proposing that KIF13B may rapidly dimerize and monomerize on microtubules? While it is not impossible, I found this to be unlikely. Later in the manuscript, the authors provide a more straightforward interpretation that the transient interactions between CAP-Gly and tubulin tail prevent full dissociation of the motor and enable reattachment of the motor domain to the microtubule. I would recommend deleting this section as it may be confusing to the reader.

We disagree that dissociation of the KIF13B Δ tail construct into its monomeric form does not likely contribute to its shorter observed run lengths. Our mass photometry data in Fig. 1 shows that the oligomeric state of this construct is more sensitive to concentration than the full-length motors. However, since we do not have direct evidence for this idea, we have removed the speculation as requested.

4. The authors wrote, “we conclude that the KIF13B’s preference for tyrosinated microtubules is nucleotide-dependent.” This can be misunderstood as if the CAP-Gly interaction with the tubulin tail is nucleotide-dependent. I recommend a revision of this sentence.

Thank you for pointing this out and we have updated our presentation of this idea to make it clearer to the reader that we are referring to the nucleotide state of the motor domain.

5. The authors also wrote that CAP-Gly binding to the tubulin tail might stimulate dimerization and activation of processive motility. The authors do not present any evidence of dynamic dimerization and monomerization of KIF13B. Therefore, the sentence is too speculative as is and should be revised or removed from the manuscript.

It is true that we do not present any data about dynamic transitions between monomeric and dimeric states in our manuscript. However, our mass photometry experiments clearly show the full-length motor, along with our new CAP-Gly domain truncation, exist predominantly as monomers at the low concentrations used in our TIRF assays (< 5nM). Despite this, we now provide brightness and two-color analysis that shows that the full-length motor, along with the truncated CAP-Gly domain, interact with microtubules predominantly as dimers even in the ADP state. As we pointed out in the text, a similar conundrum has been reported for myosin 7, which appears monomeric in solution, but dimeric when bound to actin (Liu et al. 2021). We have further added the caveat that our current data does not provide evidence of lattice-mediated dimerization, but such events may be beyond the capabilities of our optical configuration (our typical integration times are ~ 500 msec). We believe pointing out this conundrum and our speculation on a possible explanation is stimulative for the field to address this question in further studies.

6. While it is conceivable that p150 binding to EB1 at the plus end may localize the motor to the starting point of dynein-mediated transport, what could be the potential advantage of localizing a plus-end-directed KIF13B to the microtubule plus end through a similar mechanism?

This is an interesting question that warrants further investigation. Data suggest that in flies, KIF13B plays a role in spindle orientation, which could be related to cortical delivery of the motor via microtubule plus-ends (Lu and Prehoda, 2013). We mention this possibility now in the Discussion.

REVIEWERS' COMMENTS

Reviewer #2 (Remarks to the Author):

The revised version of the manuscript by Fan & McKenney is greatly improved. I found two minor interpretation issues that I would suggest addressing. Otherwise, it is ready for publication.

1. Authors use SD maps (Fig. 2) to claim “strong enhancement of binding”. I still don’t quite understand this. SD shows variance in binding, but not the binding itself. Low SD may mean no binding, or very high binding (motor never unbinds). As a minimum authors should explain how they infer “strong enhancement” in the absolute sense from an SD map, which is supposed to only show a variation.

2. In their model, authors seem to suggest that CAP-Gly domain kicks in only when both heads are in ADP state, which prevents the motor from diffusing away from the microtubule. This is of course a possibility, but I guess a possible alternative (or complimentary scenario) is that CAP-Gly binds and unbinds all the time. I don’t see how going into double ADP state could suddenly turn on CAP-Gly binding, which seems to be what authors imply. More likely, CAP-Gly can bind and unbind including when only one head is in ADP state, which is a possibility consistent with experiment on figure 3F. As tempting as it is to propose one model, I think the manuscript would benefit from openly discussing possible variations.

Reviewer #3 (Remarks to the Author):

The authors have sufficiently addressed my comments on the original manuscript and performed experiments to address my main concern about recombinant tubulin. Overall, I am supportive of the publication of this revised manuscript, but I also have a few additional suggestions.

1. Regarding my concern about microtubule binding experiments in the ADP condition, I would suggest the authors include some of the points in their response to the manuscript. In particular, it would further help the reader if they can make it sufficiently clear that kinesin motor domains can readily release ADP and bind microtubules, but in the context of an autoinhibited motor, motor domains cannot easily access the microtubule lattice, and that is why they exhibit poor microtubule landing rates.

2. More importantly, is it possible to generate a truncated kinesin construct that only lacks the CAP Gly domain (deltaCG) or contains a point mutant that prohibits CG binding to microtubules? If this construct is still autoinhibited, it would allow the authors to directly demonstrate whether the CAP-Gly domain determines where the autoinhibited motor lands onto the microtubule. This would significantly boost the impact of the main story.

Reviewer #2 (Remarks to the Author):

The revised version of the manuscript by Fan & McKenney is greatly improved. I found two minor interpretation issues that I would suggest addressing. Otherwise, it is ready for publication.

We thank the reviewer for their time and helpful suggestions for our manuscript.

1. Authors use SD maps (Fig. 2) to claim “strong enhancement of binding”. I still don’t quite understand this. SD shows variance in binding, but not the binding itself. Low SD may mean no binding, or very high binding (motor never unbinds). As a minimum authors should explain how they infer “strong enhancement” in the absolute sense from an SD map, which is supposed to only show a variation.

The reviewer is correct that the SD map shows variance in binding. If the motor never unbinds from the microtubule, then the SD map may show very low signal as the reviewer suggests. However, since the motor is labeled in our experiment, we can directly observe that this is not the case in our data, as very little motor signal is observed on microtubules that correspond to weak intensity in the SD map. Further, since the experiment is performed in ATP, static binding would indicate dead/inactive motors, which we do not observe. Therefore, we directly measure the enhancement of binding events along particular types of microtubules through the measured signal increase in the SD image. The ‘strong enhancement’ we claim is derived from these measurements, plotted in the graphs in Fig. 2B-C. As we cited in our paper, this methodology has been utilized to track the preference of kinesin-1 for microtubule subtypes in living cells (Cai et al. *PLOS Bio*, 2009, Cai et al. *J. Cell Biol.* 2007, and for dynein in vitro: McKenney et al. *EMBO J.* 2016). The methodology is described further in Cai et al. 2010, *Methods in Enzymology*. Therefore, SD analysis is a convenient and well-used method in the field. To clarify this further, we have changed the statement to “strong enhancement of transient binding events” which further describes how transient changes in pixel intensities lead to stronger signal in the SD map.

2. In their model, authors seem to suggest that CAP-Gly domain kicks in only when both heads are in ADP state, which prevents the motor from diffusing away from the microtubule. This is of course a possibility, but I guess a possible alternative (or complimentary scenario) is that CAP-Gly binds and unbinds all the time. I don’t see how going into double ADP state could suddenly turn on CAP-Gly binding, which seems to be what authors imply. More likely, CAP-Gly can bind and unbind including when only one head is in ADP state, which is a possibility consistent with experiment on figure 3F. As tempting as it is to propose one model, I think the manuscript would benefit from openly discussing possible variations.

We agree with the reviewer’s assessment that the CAP-Gly domain may be engaged with the microtubule more continuously during movement, although we have no data to

support the idea directly. This is why we performed the experiment in Fig. 3F to see if any changes in motility may be observed when the motor transitions from tyrosinated to detyrosinated microtubules. However, we did not observe any changes in motility, suggesting continuous engagement of the CAP-Gly domain is not strictly required for continual processive motility, as we stated in the results. We did not mean to imply that the nucleotide state of the motor domain may directly control the engagement of the CAP-Gly domain, but rather our data clearly shows that the affinity of the motor domain in the ATP state dominates over the CAP-Gly (Fig. 4). In our model, when both motor domains enter the ADP state, there is no longer dominance of the motor domain affinity for the microtubule over the CAP-Gly domain, and this is when binding of the CAP-Gly to tyrosinated microtubules is critical to maintain the motor's attachment to the microtubule, facilitating continuation of the processive run. We updated our model description to include a sentence clarifying this point for the reader: "*When the motor domains are in the ADP state, our data reveals that the CAP-Gly domain binding to tyrosinated microtubules dominates the overall microtubule-motor interaction (Fig. 4B). Because of this, the CAP-Gly domain may facilitate fast rebinding of the motor to the lattice, preventing diffusion away from the microtubule and termination of motility (Fig. 6, step 4).*"

Reviewer #3 (Remarks to the Author):

The authors have sufficiently addressed my comments on the original manuscript and performed experiments to address my main concern about recombinant tubulin. Overall, I am supportive of the publication of this revised manuscript, but I also have a few additional suggestions.

We thank the reviewer for their time and helpful comments to improve our manuscript.

1. Regarding my concern about microtubule binding experiments in the ADP condition, I would suggest the authors include some of the points in their response to the manuscript. In particular, it would further help the reader if they can make it sufficiently clear that kinesin motor domains can readily release ADP and bind microtubules, but in the context of an autoinhibited motor, motor domains cannot easily access the microtubule lattice, and that is why they exhibit poor microtubule landing rates.

We agree that autoinhibition hampers microtubule binding, but we point out that the kinesin motor does not readily release ADP (ADP release is the rate-limiting step in the kinesin mechanochemical cycle) until stimulated ~ 1000-fold by the interaction of the motor domain with the microtubule (Hackney, 1988, Ma and Taylor, 1997). We agree with the reviewer's conclusion that the poor microtubule landing is likely due to autoinhibition. We point out in the manuscript that our data reveal this in several places: "*The landing rate of KIF13B was more than 50-fold less than that of truncated or mutant motors (Fig. 3B), reflecting the strong autoinhibition of the wild-type motor⁴⁸*" and "*We conclude that release from autoinhibition and dimerization of the motor results in a higher affinity for both types of microtubules*" and "*In addition to potential avidity effects, the increased affinity is also*

likely due to microtubule-stimulated release of ADP from the motor domain, which is enhanced upon release from the autoinhibited state⁴⁸" and "However, it's landing rate was an order of magnitude higher than full-length motors or KIF13B^{CG}, suggesting the truncation of the tail releases the motors from autoinhibition". We feel these statements make it sufficiently clear to readers that our data support the conclusion that autoinhibition of the full-length motor hampers it's interaction with the microtubule.

2. More importantly, is it possible to generate a truncated kinesin construct that only lacks the CAP Gly domain (deltaCG) or contains a point mutant that prohibits CG binding to microtubules? If this construct is still autoinhibited, it would allow the authors to directly demonstrate whether the CAP-Gly domain determines where the autoinhibited motor lands onto the microtubule. This would significantly boost the impact of the main story.

We thank the reviewer for this suggestion. While it would be significant effort for us to generate and characterize new mutant constructs, we agree this would be informative and is a good idea for future studies. We point out that in Fig. 4B, we use the full-length WT motor in ADP, which largely negates the microtubule-binding of the motor domains, but may not affect the autoinhibition state of the motor. In this experiment, it is clear that the CAP-Gly domains directs motor landing onto tyrosinated microtubules, further suggesting that this effect may be possible even in the autoinhibited state. Ultimately, it will be useful to reveal the conformation of the CAP-Gly domain in the structure of the autoinhibited motor and our full-length constructs may be useful for structural biology efforts in the future.